# An updated assessment of past and future warming over France based on a regional observational constraint

Aurélien Ribes[1], Julien Boé[2], Saïd Qasmi[1], Brigitte Dubuisson[3], Hervé Douville[1], and Laurent Terray[2]

[1]CNRM, Université de Toulouse, Météo France, CNRS, Toulouse, France
[2]CECI, Université de Toulouse, CERFACS, CNRS, Toulouse, France
[3]Météo-France, Direction de la Climatologie et des Services Climatiques, Toulouse, France

**Correspondence:** Aurélien Ribes (aurelien.ribes@meteo.fr)

**Abstract.** Building on CMIP6 climate simulations, updated global and regional observations, and recently introduced statistical methods, we provide an updated assessment of past and future warming over France. Following the IPCC AR6 and recent global scale studies, we combine model results with observations to constrain climate change at the regional scale. Over Mainland France, the forced warming in 2020 wrt 1900-1930 is assessed to be 1.66 [1.41 to 1.90] °C, i.e., in the upper range of the CMIP6 estimates, and is almost entirely human-induced. A refined view of the seasonality of this past warming is provided through updated daily climate normals. Projected warming in response to an intermediate emission scenario is assessed to be 3.8°C (2.9 to 4.8°C) in 2100, and rises up to 6.7 [5.2 to 8.2] °C in a very high emission scenario, i.e., substantially higher than in previous ensembles of global and regional simulations. Winter and summer warming are expected to be about 15% lower than, and 30% higher than the annual mean warming, respectively, for all scenarios and time periods. This work highlights the importance of combining various lines of evidence, including model and observed data, to deliver the most reliable climate information. This refined regional assessment can feed adaptation planning for a range of activities and provides additional rationale for urgent climate action. Code is made available to facilitate replication over other areas or political entities.

## 1 Introduction

The 6th Assessment Report of the International Panel on Climate Change (IPCC, 2021, hereafter IPCC AR6) has recently provided an up-to-date assessment of the current knowledge on past and future climate change. This new report builds on improved physical understanding, updated observations, a new generation of Earth System Models (ESMs), and a wide range of published methodologies and results to deliver the latest expectation about future climate change.

In the IPCC AR6, a particular effort was made to provide regional scale information on observed and projected changes, including an interactive atlas (Gutiérrez et al., 2021) which provides an assessment of recent and future changes in simple climate indices aggregated at a subcontinental scale. One reason given for this regional focus is that "The impacts of climate change are generally experienced at local, national, and regional scales, and these are also the scales at which decisions are typically made." (Arias et al., 2021, TS.1.4). Yet, the IPCC AR6 regions are still large supranational domains, while decision makers are mostly interested in national (e.g., https://www.cmcc.it/g20, Soubeyroux et al., 2021) or even smaller scale studies (e.g., http://www.acclimaterra.fr/, https://reco-occitanie.org/crocc_2021/).

Here, we seek to provide an assessment of past and future climate change at the scale of Mainland France, with a particular focus on projected mean climate change up to 2100. This assessment could directly feed and benefit impact studies, adaptation planning, and mitigation policies at the national level. It is mostly based on CMIP6 global projections, although results will be compared to those of the latest EURO-CORDEX ensemble of regional climate models (Jacob et al., 2014). While the IPCC AR6 recognizes that higher-resolution limited-area models add value in representing many regional weather and climate phenomena, especially over regions of complex orography, their fit-for-purpose for future projections heavily depends on key processes, forcings and drivers which are not necessarily better represented than in global Earth System Models (e.g. Doblas-Reyes et al., 2021; Boé et al., 2020a).

An important novelty of the IPCC AR6, if compared to the IPCC AR5 (IPCC, 2013), is the use of observational constraints for generating 21st century projections. Unlike previous IPCC assessment reports, projections of global mean surface air temperature (GSAT) were not derived directly from the raw results of all available ESM simulations. Instead, model simulations were used in combination with historical GSAT observations to derive future warming ranges consistent with the observed record. This approach was supported by various studies showing consistent results and added value, in particular an overall reduction in the intermodel spread compared to raw model results (Brunner et al., 2020; Nijsse et al., 2020; Tokarska et al., 2020; Liang et al., 2020; Ribes et al., 2021). However, while the IPCC AR6 provides constrained projections for several global mean variables (near-surface temperature, OHC, sea level), constrained projections at the regional scale were not available, introducing a possible source of inconsistency between global and regional assessments.

To circumvent this problem, the notion of "global warming level" (GWL) was used. In this way, the spatial distribution of the expected warming was described for various levels of GSAT warming, e.g., +1.5°C, +2°C or +3°C above pre-industrial (1850-1900). This approach has at least two drawbacks. First, the uncertainties on the spatial pattern of warming (e.g., Lopez et al., 2014; Zappa et al., 2020) come on top of those related to the GWL, in such a way that it is difficult to deduce uncertainty ranges on the expected local warming at a given 21st century period. Second, this approach only uses the GSAT observational constraint, and the additional information provided by local or regional observations is not taken into account.

In this study, we overcome these issues by providing constrained temperature projections at the regional scale, which account for both global mean and regional temperature observations. We also apply recently introduced statistical methodologies to provide an updated and refined picture of past, present, and future climate change over France. This includes an assessment of attributable past warming and warming rate, an estimation of today's daily climate normals, and a range of constrained projections, with uncertainties, for various emission scenarios. Our assessment deals with mean temperature at the annual and seasonal scale. Changes in annual and seasonal precipitation are briefly discussed in SI, but no observational constraint is applied in that case. We do not assess other variables nor changes in extreme events of temperature or precipitation.

Previous academic studies, as well as non-academic reports, have addressed the question of on-going climate change over France in the last decade. Terray and Boé (2013) provide an assessment of mean climate change based on global CMIP5 models. Various national reports (Peings et al., 2011a, b; Ouzeau et al., 2014; Soubeyroux et al., 2021) also discussed past and future mean climate change based on various ensembles of global and regional climate model simulations, including bias-corrected CORDEX simulations. However, none of these used the latest CMIP6 generation of ESMs, considered observational

constraints, nor attempted to attribute recent changes to specific external forcings. This study also provides an up-to-date assessment based on latest observations, including the latest and warmest decade ever recorded since around 6500 years ago (IPCC, 2021).

Given the importance of the local and national scale in decision making, adaptation planning and mitigation policy, we expect this assessment to be of high interest for the national community and stakeholders. We also believe that this paper

provides an easily reproducible example of study and diagnostics that can be conducted to quantify, characterise and monitor climate change over a region of interest, especially where reliable and homogenized multi-decadal observations are available. The codes and data used in this work are provided to facilitate replication over any other area of interest.

## 2 Data and Methods

### 2.1 Model data

We consider an ensemble of CMIP6 models summarized in Appendix A1. For each model, we consider historical and multiple scenario (SSP1-2.6, SSP2-4.5, SSP3-7.0, SSP5-8.5; O'Neill et al., 2016) simulations, and compute the model mean over all available members. Data from the native model grid are then interpolated into a regular 0.1° grid (i.e., about 10km resolution), using a nearest neighbour interpolation accounting for the sea-land mask. In practice, any land grid-point from the target grid (points of interest are over land) takes the value of the closest land grid point from the source grid. A particular grid-point

is considered to be a land point if its land fraction is higher than 75%. Next, we compute the spatial average over France, resulting in a univariate monthly time-series. Annual and seasonal means are finally derived. Temperature and precipitation are processed in the same way.

In order to provide attribution statements, we also use hist-GHG simulations (i.e., simulations where GHGs follow their historical concentrations, but other forcing agents are kept constant; DAMIP, Gillett et al., 2016). As many CMIP6 models

have not performed hist-GHG experiments (18 out of 27 models in this study), their response to GHG-only is reconstructed (i.e., inferred) from the 1%-$CO_2$ experiment. For GSAT, this reconstruction is made as in Ribes et al. (2021, see their Supplementary Material 1.4). For regional mean temperature, we assume that pattern scaling applies (e.g., Tebaldi and Arblaster, 2014) despite some aforementioned limitations, and derive the time series of regional GHG-induced warming as the GSAT time-series multiplied by a regional scaling factor. This scaling factor is estimated as the regional to global warming ratio in

the 1%-$CO_2$ experiment (considering the first 20 years vs last 20 years of this 140yr-long experiment).

Lastly, previous ensembles of climate models are used to provide a historical perspective on our results. In particular, we consider a set of CMIP5 climate models (Appendix A2), and a set of EURO-CORDEX regional climate models (RCMs, Appendix A3). EURO-CORDEX is an ensemble of RCMs driven by lateral boundary conditions from CMIP5 global models (Jacob et al., 2014). It involves a higher spatial resolution than CMIP5 models, and is therefore often used for adaptation

planning.

## 2.2 Observational data

To characterize the past warming over Mainland France, we use data from the National Thermal Index (ITh). This index is obtained by averaging data from 30 measurement stations, well distributed over the country. For each station time series, monthly measurements are homogenized following a state-of-the art pairwise method (Mestre et al., 2013), that was applied to a much larger set of measurement stations. In addition to being spatially representative, the 30 stations used in ITh are selected as to provide data since 1900 at least, and limited homogeneity breaks after 1947. As a result, before 1947, the ITh index is constructed as the average of the homogenized monthly data – it is available since 1899. After 1947, daily (unhomogenized) values are used, and the ITh index is available at the daily time-step. Monthly ITh values are calculated as the monthly average of daily values, and are well consistent with the homogenized series. All 30 measurement stations are located at low-altitude. There is no estimate of measurement uncertainty provided with this product.

In this study, we use this national index rather than more common global datasets such as CRUTEM (Osborn et al., 2021) or BEST (Rohde and Hausfather, 2020), for two reasons. First, it is available both at the monthly and daily time-scale. Second, the pairwise homogenization procedure is applied to a large sample of raw temperature data, which should make this procedure more accurate than in other datasets. However, a simple comparison with a reconstruction of the average France temperature from the CRUTEM5 dataset suggests that the two datasets agree very well.

Our observational constraint procedure also requires GSAT observations. We use the HadCRUT5 dataset for GSAT observations since 1850 (Morice et al., 2021). The corresponding ensemble is used to assess observational uncertainty.

## 2.3 Statistical methods

This study makes use of various statistical methods which have been previously introduced and evaluated in the literature. Here, we only review the key concepts of these techniques, and discuss the choices that have been made in implementing these techniques. A full description is available in the corresponding papers.

### 2.3.1 Observational constraints and attribution

A key novelty of this study is to assess past and future climate change using an observational constraint method that has been previously applied to global mean warming (Ribes et al., 2021) and local or regional warming (Qasmi and Ribes, 2021). This technique is called Kriging for Climate Change (KCC) and works in 3 steps. First, the forced response of each climate model considered is estimated over the period 1850-2100. In order to also get attribution statements, the responses to ALL (all forcings), NAT (natural forcings only) and GHG forcings are estimated separately. Second, the sample of forced responses from available CMIP6 climate models is used as a prior of the real-world forced response. Third, observations are used to derive a posterior distribution of the past and future forced response given observations, in a Bayesian way.

The procedure can be summarised using the following equations:

$$\mathbf{y} = \boldsymbol{H}\mathbf{x} + \boldsymbol{\varepsilon}, \tag{1}$$

where $\mathbf{y}$ stands for observations (a vector), $\mathbf{x}$ stands for the forced response (a vector), $\boldsymbol{H}$ is an observational operator (matrix), $\boldsymbol{\varepsilon}$ is the random noise associated with internal variability and measurement errors (a vector), and $\boldsymbol{\varepsilon} \sim N(\mathbf{0}, \boldsymbol{\Sigma}_{\mathbf{y}})$, where $N$ stands for the multivariate Gaussian distribution. The observational operator $\boldsymbol{H}$ is typically a very simple matrix which extracts the components of $\mathbf{x}$ that are observed in $\mathbf{y}$, i.e., all entries are equal to 0 or 1 (see Appendix B). Climate models are used to construct a prior on $\mathbf{x}$: $\pi(\mathbf{x}) = N(\boldsymbol{\mu}_{\mathbf{x}}, \boldsymbol{\Sigma}_{\mathbf{x}})$. Then the posterior distribution given observations $\mathbf{y}$ can be derived as $p(\mathbf{x}|\mathbf{y}) = N(\boldsymbol{\mu}_p, \boldsymbol{\Sigma}_p)$. Remarkably, $\boldsymbol{\mu}_p$ and $\boldsymbol{\Sigma}_p$ are available in closed-form expressions.

In the following, we are interested in assessing the forced response of annual, summer and winter mean temperature over France (projections), as well as the annual mean response to specific subsets of forcings (attribution). These forced responses could be constrained by various observations. Here, we consider constraints by GSAT observations, and by regional (i.e., averaged over France) annual mean temperature only – the rationale behind this choice is discussed below. Therefore,

$$\mathbf{x} = \left( \mathbf{T}_{\mathrm{glo}}^{\mathrm{all}}, \mathbf{T}_{\mathrm{ann}}^{\mathrm{all}}, \mathbf{T}_{\mathrm{ann}}^{\mathrm{ghg}}, \mathbf{T}_{\mathrm{ann}}^{\mathrm{nat}}, \mathbf{T}_{\mathrm{jja}}^{\mathrm{all}}, \mathbf{T}_{\mathrm{djf}}^{\mathrm{all}} \right), \tag{2}$$

where each element is a time-series of the forced response over the period 1850–2100 (except $\mathbf{T}_{\mathrm{ann}}^{\mathrm{ghg}}$ which only covers 1850–2020), $\mathbf{T}$ stands for temperature, 'all', 'ghg' or 'nat' are the subsets of external forcings considered, 'glo' means GSAT (regional temperature is considered where 'glo' is not written), and 'ann', 'jja' or 'djf' are the annual mean, summer mean, and winter mean, respectively. As a result, the length of $\mathbf{x}$ is $n_{\mathbf{x}} = 1426$. Similarly,

$$\mathbf{y} = \left( \mathbf{T}_{\mathrm{glo}}^{\mathrm{obs}}, \mathbf{T}_{\mathrm{reg}}^{\mathrm{obs}} \right), \tag{3}$$

i.e., only observed time-series are used in $\mathbf{y}$. The length of these time-series varies: 1850-2020 for GSAT, 1899-2020 for the French ITh dataset, leading to a total length of $n_{\mathbf{y}} = 293$ for $\mathbf{y}$. Finally, all attribution and projection diagnoses presented below can be derived from the posterior distribution $p(\mathbf{x}|\mathbf{y})$.

Accounting for GSAT is important because various recent studies argued that the observational constraint on this variable is robust (e.g., to the choice of the statistical method), with the high-end of simulated GSAT model responses not consistent with observed GSAT changes (Lee et al., 2021, and references therein). As there is a clear correlation (across CMIP models) between GSAT and local warming over most regions including France, a reduced GSAT response is expected to imply a reduced regional warming. This is confirmed by Qasmi and Ribes (2021), who found that accounting for the global constraint clearly improves the accuracy of local projections. Accounting for local observations is also attractive, especially over regions where long observational records are available and the climate change signal has already emerged. Qasmi and Ribes (2021) also report a significant added-value in doing so, although limited given the modest regional signal-to-noise ratio.

The data to be included in the observational constraint represent a key element of the proposed method which is further discussed in Section 3. We do not consider constraints by observed seasonal temperatures, as this would involve additional technical challenges (e.g., to model statistically the dependence between annual and seasonal means), and preliminary tests suggest that there is no clear added-value in doing so.

The choice of the prior is another key element of this approach. Here, we consider the entire CMIP6 ensemble, assuming that all models are equally plausible, and that "models are statistically indistinguishable from the truth". Our choice is in

contradiction with Hausfather et al. (2022), who suggest that the most sensitive (to atmospheric $CO_2$) CMIP6 models should be excluded from projections, as evidence suggest that these models are too sensitive. Indeed, the IPCC AR6 provided GSAT projection ranges accounting for observational constraints (Lee et al., 2021) which, in the end, excluded the most sensitive models. However, evidence that these models are too sensitive comes primarily from GSAT historical observations. In this study GSAT observations are taken into account in $\mathbf{y}$, in a way consistent with Ribes et al. (2021) and therefore Lee et al.

(2021). There would be a risk of circular reasoning in modifying the prior to exclude "hot-models", since information from observations $\mathbf{y}$ would be used to design the prior $\pi(\mathbf{x})$, which is not a good practice in Bayesian statistics. Instead, the prior $\pi(\mathbf{x})$ is expected to be representative of available knowledge before accounting for observational evidence. We assume that model democracy remains a reasonable choice in this context.

Implementing this methodology requires determining the values of $\boldsymbol{\mu_x}$, $\boldsymbol{\Sigma_x}$, and $\boldsymbol{\Sigma_y}$. Following Ribes et al. (2021), $\boldsymbol{\mu_x}$ and

165 $\boldsymbol{\Sigma_x}$ are estimated as the sample mean and covariance of the CMIP6 model forced responses. Estimating $\boldsymbol{\Sigma_y}$ requires statistical modelling of internal variability and measurement error. Note that *measurement error* is meant in a broad sense here, as it encompasses all errors involved in estimating a global or regional temperature average, including individual measurement error, but also the treatment of incomplete data coverage, homogenization uncertainty, and others. Regarding GSAT, we follow Ribes et al. (2021) in using a sum of two Auto-Regressive processes of order 1 (AR1) to model internal variability, and

170 the HadCRUT5 ensemble to estimate measurement uncertainty. Regarding annual mean temperature over France, we follow Ribes et al. (2016) in assuming that internal variability follows an AR1($\alpha$=0.2) process. We assume no measurement error in regional temperature observations. We provide further details and discussion in Appendix B about the structure of $\boldsymbol{H}$, and about estimation of the inputs parameters $\boldsymbol{\Sigma_x}$ and $\boldsymbol{\Sigma_y}$. As a final remark, the current implementation does not account for uncertainty in the input parameters $\boldsymbol{\mu_x}$, $\boldsymbol{\Sigma_x}$, and $\boldsymbol{\Sigma_y}$ – this could be done, e.g., using more complex hierarchical Bayesian

models.

### 2.3.2 Climate normals

Another way to characterize the observed climate change to date is to estimate up-to-date climate normals. Climate normals are routinely computed by National Weather Services (NWSs), and are used as a baseline to describe the mean temperature that can be expected for a given day or month. The World Meteorological Organization (WMO) provides guidance for calculating

these normals as an average over 3 decades. In a warming climate, however, such normals lag behind the current climate. Rigal et al. (2019) discussed this issue and proposed a statistical method to estimate up-to-date climate normals at the daily resolution.

The proposed technique can be summarized as follows. It is assumed that

$$y_{d,y} = f(d) + g(y).h(d) + \varepsilon_{d,y}, \tag{4}$$

where $y_{d,y}$ is the observed temperature for day $d$ and year $y$, $f(d)$ is the climate normal for day $d$ without any climate change, $g(y).h(d)$ describes the impact of climate change on climate normals, and $\varepsilon_{d,y}$ denotes internal variability. This statistical model basically assumes that (i) the human-induced perturbation can be factorized as $g(y).h(d)$, and (ii) the functions $f()$,

$g()$ and $h()$ are smooth. Rigal et al. (2019) then propose an algorithm to estimate $f()$, $g()$ and $h()$ directly from a set of daily observations $y_{d,y}$, using smoothing splines.

Here, we use a similar methodology to estimate non-stationary climate normals over the period 1947–2020. Beyond providing a useful illustration of daily changes, this enables us to investigate how exactly the estimated forced warming is distributed throughout the year. To do this, we adapt the procedure of Rigal et al. (2019). Instead of estimating $g()$ from observations only, we take it from the observational constraint method described above, to ensure consistency among results. Estimation of $f()$ and $h()$ is even simpler in that case: for each day $d$, $(y_{d,y})$ are regressed onto $g(y)$, and the resulting regression coefficients are smoothed. Lastly, the estimate of $h()$ is rescaled so that its annual mean value is 1. This ensures consistency between the warming over the entire period and the assumed $g()$. One important step in implementing this methodology is the selection of the number of degrees of freedom for each function. Here, we follow recommendations from Rigal et al. (2019), and use $df_f = 15$ for $f()$ and $df_h = 6$ for $h()$.

Another methodological novelty concerns the estimation of confidence intervals around non-stationary normals, through a resampling technique. First, we sample uncertainty on $g()$ by sampling the posterior of our observational constraint (i.e., we draw perturbed estimates of $g()$). Second, we bootstrap observations over years (i.e., random re-sampling with replacement of $n$ years among the $n$ available years), to derive resampled estimates of $f()$ and $h()$. Uncertainty ranges are finally derived as empirical 5–95% quantiles of these perturbed estimates.

## 3   Results

### 3.1   Warming to date

First, we assess the past warming to date, and discuss various estimation techniques that can be used. This includes in particular a simple smoothing spline technique (purely observational estimate), the raw CMIP6 multi-model mean (purely model estimate), and several variants of the KCC method, which combine information from models and observations (Figure 1).

The smoothing technique, which does not account for any model information, suggests a warming of nearly +2°C in 2020 with respect to 1900-1930 (all estimates within this subsection are meant with respect to the 1900-1930 baseline). However, several record-breaking values observed over the latest years may contribute to inflate this value. The raw CMIP6 multi-model mean leads to a much lower value of about +1.44°C in 2020. Interestingly, over the last 20-yr, only 3 values for individual years fall below the forced response simulated by CMIP6 models – suggesting that this estimate might be biased low. A potential underestimation is also supported by the fact that, on average over the last 20-yr, the CMIP6 multi-model mean has warmed by 0.4°C less than observations (+1.14°C vs +1.53°C, respectively). This suggests that the case of France is quite specific: regional observations suggest that the CMIP6 multi-model mean historical warming is underestimated over France, while no such founding was made globally, and accounting for GSAT observations typically leads to revising CMIP6 projections downwards (Tokarska et al., 2020; Ribes et al., 2021). This finding has key implications in terms of the observational constraints that may be applied regionally. Consistent with global scale results, accounting for GSAT observations in the constraint tends to revise CMIP6 ranges downwards over France (like almost every region). In contrast, accounting for regional observations in

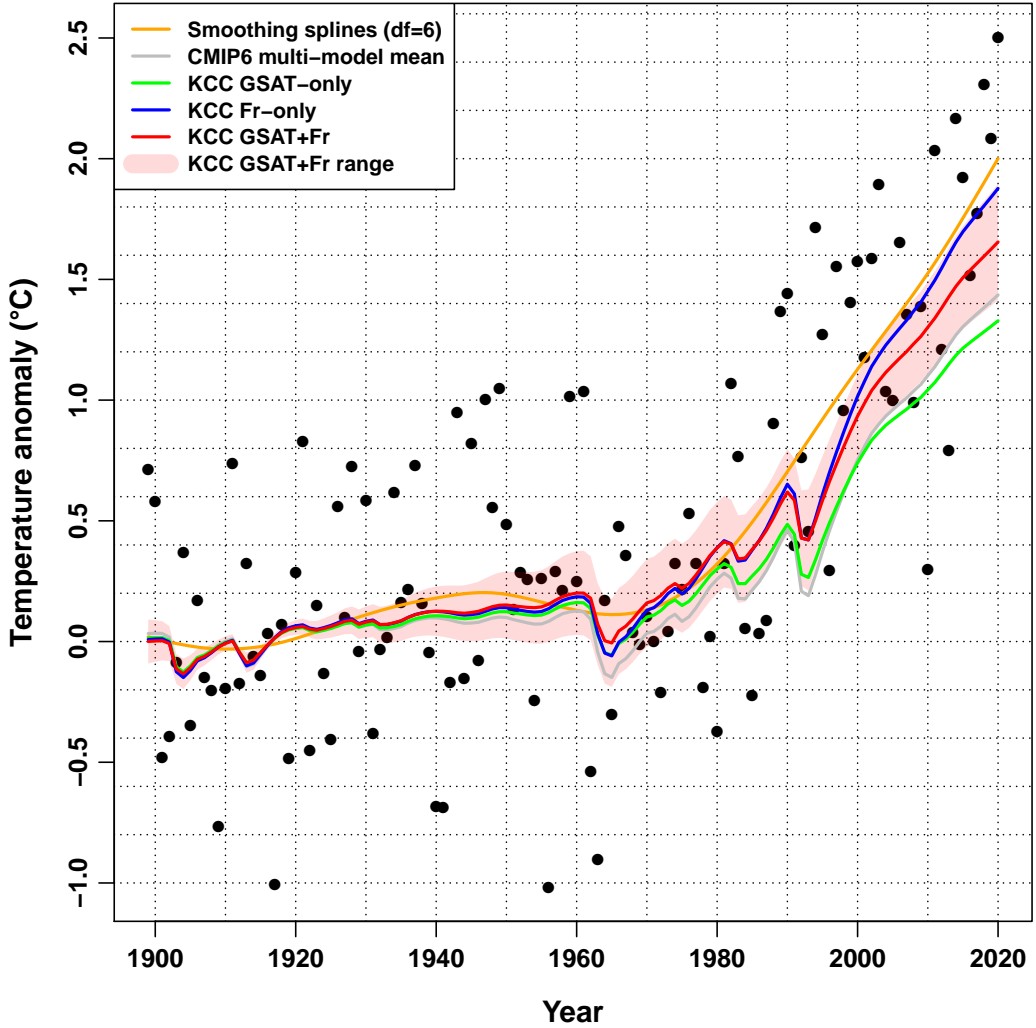

**Figure 1. Observations vs forced response estimates.** Observed annual mean temperature over France (1899–2020, black points) are compared to various estimates of the forced response over the same period. Orange: simple smoothing spline estimate, using df=6 (degrees of freedom). Grey: CMIP6 multi-model mean estimate (best-estimate only). Green: Result of the KCC constraint using only GSAT observations (best-estimate only). Blue: Result of the KCC constraint using only regional observations (i.e., over France). Red: Result of the KCC constraint using both GSAT and regional observations to build the constraint. The 5–95% uncertainty range assessed in the latter case.

the constraint tends to revise CMIP6 ranges upwards over our area of interest. Finally, these two sources of observations have competing effects, and their respective strengths have to be examined carefully.

Three variants of the KCC constraint illustrate this point (Figure 1). If only GSAT observations are used, the estimated forced warming is revised downwards compared to the raw CMIP6 multi-model mean, at +1.33°C in 2020. If only observations over France are used, the estimated forced warming is widely revised upwards, to +1.88°C in 2020. This value is close to the smoothing splines estimate, although the shape of the time-series is different – the KCC result is clearly constrained by knowledge about external forcing time-series including, e.g., volcanic eruptions. Lastly, if the two sources of observations (i.e., GSAT and France) are considered simultaneously in the constraint, they partly cancel each other out, leading to an intermediate estimate of +1.66°C in 2020. These various estimates suggest that methodological choices about estimating forced warming to date play a larger role regionally rather than globally, due to the higher role of internal variability, and corresponding smaller signal-to-noise ratio, at the regional scale.

Among these various estimates which one should be preferred? Until recently, studies estimating past warming often considered purely observational estimates, while most projections focused only on model results – leading to potential inconsistencies. Here, we try to merge the two sources of information to provide a consistent view on past and future changes. By design, considering only regional observations leads to a better visual fit with the observed historical warming. However, considering the two sources of information seems desirable to take full advantage of available data, and to ensure consistency across spatial scales. Importantly, the analysis of the accuracy of these options in a perfect model framework (Qasmi and Ribes, 2021) suggests that there is strong added-value in considering GSAT observations to compute regional scale projections, and that constraining by GSAT and regional observations (GSAT+reg) leads to the highest score. For this reason, we think that the combined GSAT+reg option should be favoured to assess both past and future warming. Restriction may apply to this choice if all CMIP models simulate a wrong relationship between GSAT and regional temperature (e.g., a biased regional warming ratio, in which case the GSAT information may not be considered in the constraint).

Is there evidence that the range of CMIP models does not capture the correct regional to global warming ratio? To investigate this question, we focus on the global and regional observed warming over the last 20 year (2001-2020 wrt 1900-1930, i.e., a period over which the gap between raw CMIP6 simulations and observations is relatively large; Figure 2). Observations over this 20yr-period are not representative of the forced response only, as they are also affected by internal variability, especially regionally. However, our analysis suggests that they fall within the range of CMIP6 responses. Looking at the regional to global warming ratio specifically, most CMIP6 models simulate a value between 1 and 1.25, and the 90% range implied by the CMIP6 ensemble is [.85, 1.51] (median at 1.18). The observed warming ratio over the last 20yr, 1.49, falls within this range, although barely. Taking a symmetrical point of view, we can assess a confidence range around the observed value of 1.49, i.e., by quantifying how much internal variability over such a 20yr period could affect the observed value. The resulting confidence range of [1.17, 1.90] suggests that observations are consistent with a warming ratio of 1.2, i.e., a value close to the CMIP6 median. Furthermore, the KCC method (GSAT+reg) successfully narrows the uncertainty on the warming ratio (and the forced response in general). After applying the KCC constraint, the assessed warming ratio 90% confidence range becomes [1.16, 1.56] (median: 1.36), which is well consistent with observations. Similar findings are made considering the last decade

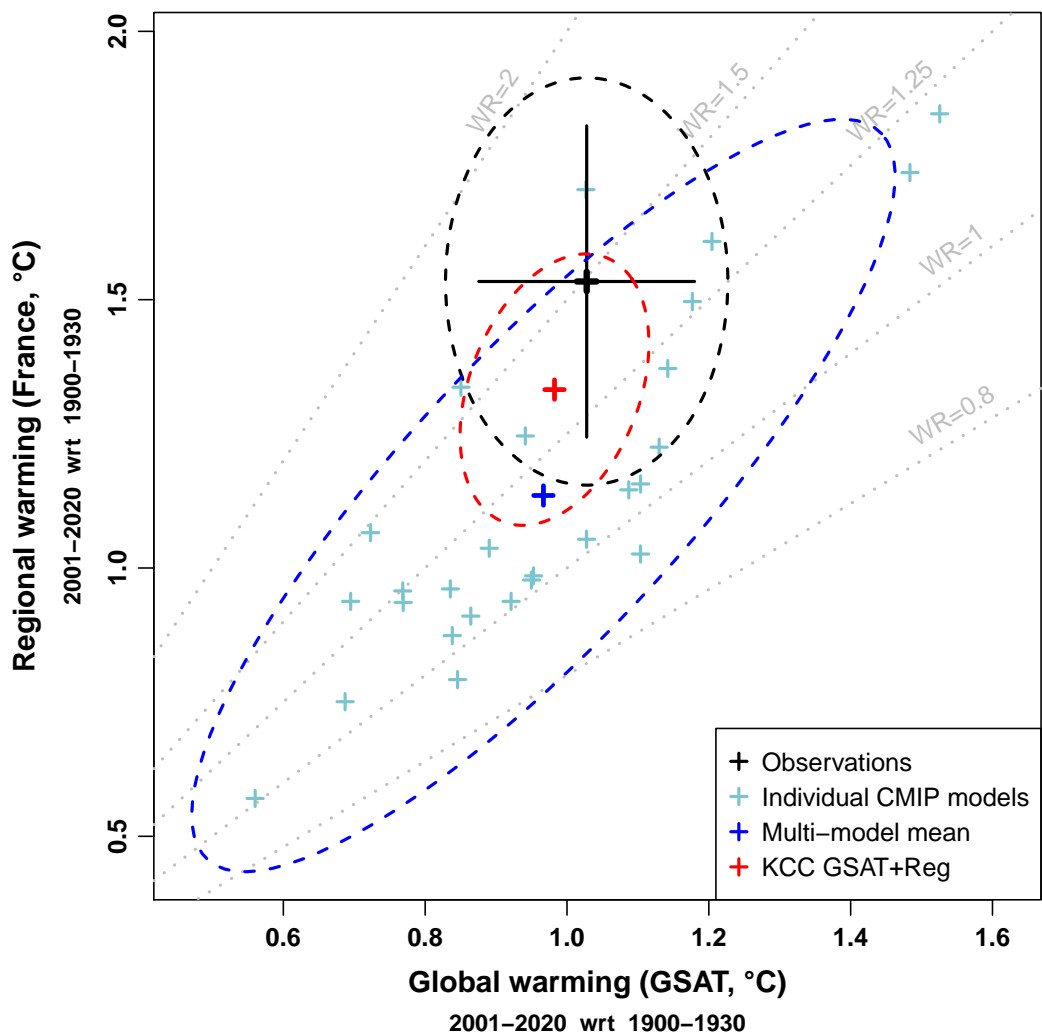

**Figure 2. Global to regional warming over the period 2001-2020.** Comparison of the global (GSAT) and regional (France) warmings, over the period 2001–2020 with respect to 1900-1930. Black: observations (cross), 5–95% confidence ranges for the global and regional warmings separately (large cross), and 90% confidence 2-D region (dashed ellipse). Confidence ranges and region are based on assumed internal variability and observational uncertainty at global and regional levels. Light blue: estimated forced response for each CMIP6 model individually. Blue: CMIP6 multi-model mean (cross) and 90% confidence region (dashed ellipse), derived from the model spread. This confidence region corresponds to the prior used in the KCC constraint. Red: KCC constrained estimate using both GSAT and regional observations to build the constraint. Grey: Oblique dotted lines show regional to global warming ratio (WR) of 0.8, 1, 1.25, 1.5 and 2.

(2011-2020 wrt 1930-1900, Figure S1), although (i) internal variability plays even a stronger role over such a short time period, and (ii) evidence suggests that internal variability contributed to make the last decade particularly hot over France. Overall, we find no clear evidence that the CMIP6 ensemble is biased low in terms of expected regional to global warming ratio over France. Instead, results from the KCC constraint are found to narrow uncertainty on recent warming relatively efficiently, while staying consistent with both models and observations. We therefore use results from this technique to estimate the amount of recent forced warming over France.

Finally, we assess the forced warming in 2020 (wrt 1900-1930) to be 1.66 [1.41 to 1.90] °C. The assessed lower bound is close to the unconstrained CMIP6 multimodel mean, suggesting that models simulating a lower recent warming are inconsistent with the observed historical warming. The assessed upper bound is close to the outcome of the KCC Fr-only constraint (i.e., applying the KCC constraint with regional observations only, ignoring the GSAT observations). The assessed forced warming over the last decade (2011-2020 wrt 1900-1930) is 1.51+/- 0.22°C. As this decade was 1.83°C warmer than the 1900-1930 baseline, our assessment implies that internal variability contributed to make that particular decade hotter than expected, by as much as 0.32°C. Our method can also be used to assess the forced warming with respect to the 1850-1900 baseline, consistent with the IPCC AR6, although no observation is available prior to 1899. Comparing the decade 2010-2019 to 1850-1900, we find a forced warming of 1.46 [1.21 to 1.70] °C, close to (and slightly lower than) the average land warming of 1.59 [1.34 to 1.83] °C, reported in the IPCC AR6 (Gulev et al., 2021). The forced warming in 2020 wrt 1850-1900 is assessed to be 1.63 [1.36 to 1.91] °C. This result underlines how limited forced changes were prior to the 1900-1930. In fact, external variability is found to play a relatively modest role over a much longer period, up to 1980, as the forced warming estimates shown in Figure 1 agree on a limited temperature change prior to this date. Causes for this lack of warming are revealed by the attribution analysis below.

## 3.2 Attribution to different forcing agents

Attributing past warming to various subsets of external forcings or individual forcing agents is an important step for understanding recent observed changes. Attribution statements were central in previous IPCC assessment reports, in particular regarding the anthropogenic forcings (ANT) vs natural forcings (NAT) contributions. However, estimating the greenhouse gases (GHGs) and other anthropogenic (OA; a subset including all non-GHG anthropogenic forcings, usually dominated by aerosols) contributions is far less common at the regional scale. Disentangling these two contributions is particularly challenging based on fingerprinting techniques, due to collinearity issues (Ribes and Terray, 2013; Jones et al., 2016). New techniques such as the one used in this study makes this assessment easier. However, beyond the choice of the statistical method, various issues still make this attribution a challenging exercise: the limited number of models participating in DAMIP (Gillett et al., 2016, this implies that GHG-only experiments are missing for some models and have to be reconstructed), the limited number of members in single-forcing attribution simulations, and the difficulty for accurately estimating regional scale forced responses in these simulations due to regional internal variability. For these reasons, the attribution results presented hereafter may be less robust than the estimates of past or future (total) forced warming.

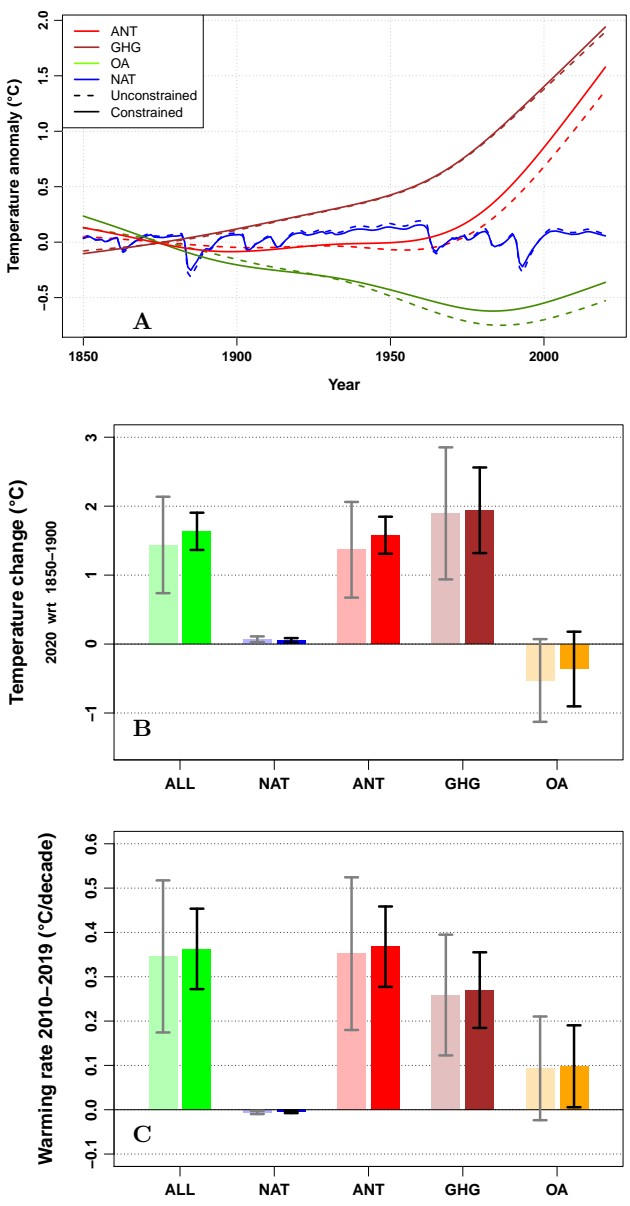

**Figure 3. Attribution of warming to date and warming rate.** (A) Constrained and unconstrained time-series of the response to Natural (NAT), Greenhouse Gases (GHG), Other Anthropogenic (OA), and Anthropogenic (ANT = GHG + OA) forcings over the period 1850-2020. (B) Temperature change induced by various subsets of external forcings over the historical period [estimated in 2020 with respect to (wrt) 1850–1900]. For each subset of forcings, the left hand-side bar and gray confidence interval describe the unconstrained CMIP6 model range, assuming a Gaussian distribution. The right hand-side bar and black confidence interval correspond to results constrained by global and local observations. All ranges shown are 5 to 95% confidence ranges. The SSP2-4.5 scenario is used to extend historical simulations after 2014. (C) Same analysis for the 2010–2019 warming rate, computed as a linear trend over that period and expressed in degrees Celsius by decade.

In this subsection, we assess the contributions of specific forcing agents to France-averaged temperature change with respect to the 1850-1900 baseline period, consistent with IPCC AR6. The rationale behind this choice is as follows. Unlike the total forced contribution (ALL), model experiments suggest that both GHGs and OA have induced noticeable temperature changes as early as the late 19th century or early 20th century (Figure 3A). In this respect, considering a preindustrial baseline avoids finding an aerosols-induced warming resulting from today atmospheric concentrations being lower than over the baseline.

We find that the regional warming to date since the pre-industrial (i.e., 2020 wrt 1850-1900) of 1.63 [1.36 to 1.91] °C is almost entirely due to the human influence (ANT), of 1.58 [1.31 to 1.85] °C (Figure 3B). The natural forcings are assessed to have a very small contribution of 0.06 [0.03 to 0.09] °C. The GHG-induced warming is assessed to be 1.94 [1.32 to 2.56] °C, partly offset by a cooling induced by other anthropogenic forcings of -0.36 [-0.90 to +0.18] °C, among which aerosols play a dominant role. Noticeably, the uncertainty on the GHG and OA contributions is larger than that on the total ANT or ALL responses.

Figure 3A reveals that the GHG and OA contributions canceled each other remarkably well prior to 1970. Both contributions explain about 0.5°C of mean temperature change over that period, leading to a small ANT signal. Several volcanic eruptions occurring in the second half of the 20th century also contributed to partly offset the GHG-induced warming, keeping the ALL warming below +0.5°C until the very late 20th century. The partial recovery from the aerosol cooling (over the last 40-yr) and the recovery from volcanic induced cooling then contributed to a very rapid warming over the last 30-yr. This combination of forcings explains why Western Europe as a whole has experienced a very abrupt warming over the last decades, while showing little or no sign of a changing climate previously (e.g., Sippel et al., 2020).

The analysis of external forcing contributions to the 2010-2019 warming rate (Figure 3C) suggests that the current warming rate is 0.36 [0.27 to 0.45] °C/decade – meaning +0.1°C of warming every 3 years. Again, this trend is assessed to be entirely human-induced, as the contribution from natural forcings is very small. However, we warn that the exact quantification of NAT-induced warming rate over this particular period is sensitive to the assumed NAT forcings in SSP scenarios (SSP2-4.5 is used after 2014 in CMIP6 historical simulations, as observed NAT forcing time-series were not available at the time simulations were made), and should be taken with caution. Interestingly, the OA-induced trend (mostly reflecting a regional decrease of anthropogenic aerosol emissions) is responsible for a warming rate of 0.1 [0.01 to 0.19] °C/decade, i.e., more than one quarter of the current warming rate.

## 3.3 Climate normals

Application of the Rigal et al. (2019) method to daily mean temperature observations since 1947 provides an estimate of changing climate normals (Figure 4). As a preliminary remark, application of the original Rigal et al. (2019) method, that is basically a smoothing technique, leads to an annual mean warming estimate consistent with the smoothing spline estimate shown in Figure 1, i.e., a value different from the one found using KCC. Here, this warming estimate is rescaled to make it consistent with our assessed forced warming. In this way, we constrain the mean warming between 1947 and 2020 to be 1.49°C. The difference with the number given in 3.1 comes from the change in the reference period (1947 is used as a baseline here, instead of 1900-1930).

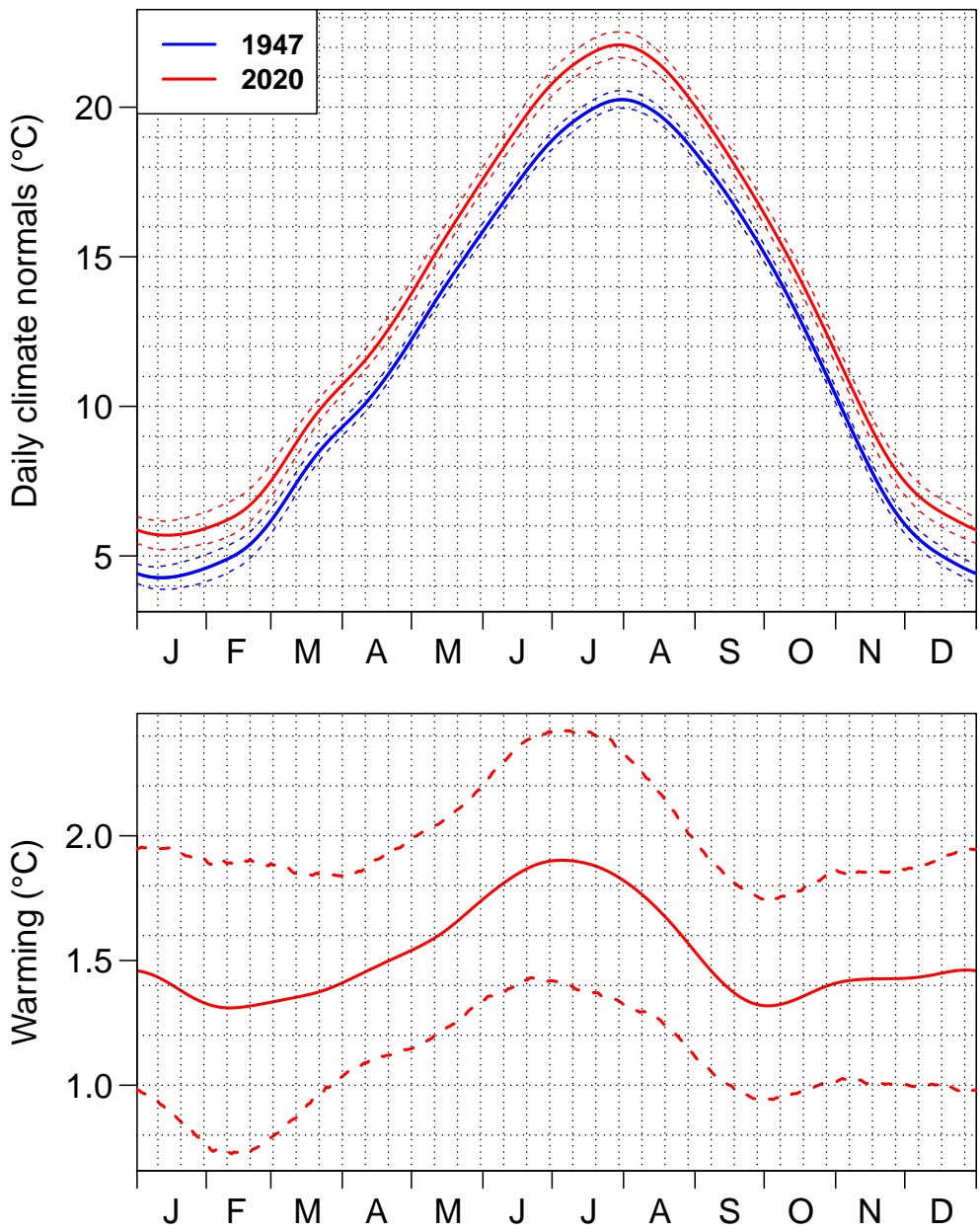

**Figure 4. Changing climate normals** Top: Daily climate normals for the daily mean temperature over France, estimated in 1947 and 2020, following the methodology of Rigal et al. (2018). Bottom: Difference between the 1947 and 2020 climate normals, i.e., 1947 to 2020 warming as function of the time of the year. Dashed lines provide bootstrap 5–95% confidence ranges (top and bottom).

This analysis reveals that the observed warming exhibits some seasonal variations. Winter and fall are subject to less warming, with a best-estimate warming typically around 1.4°C. In contrast, summer has experienced a strengthened warming of about 1.8°C, which peaks around July 1st at about 1.9°C. These results suggest some summer warming amplification over this region, that is also seen by models. Application of our technique provides a purely observational estimate of the summer to winter warming ratio of about 1.3. This value is consistent with model estimates as discussed below. However, uncertainty analysis suggests that uncertainty in the amount of warming that occurred for a given day is relatively large, typically $\pm 0.5$°C. As a consequence, the summer to winter warming ratio is poorly constrained by observations, and the closeness with model results (including projections) seems partly coincidental.

The direct comparison of daily normals in 1947 vs 2020 also gives clear indication of how the seasonal clock is affected by climate change. For instance, spring temperatures have shifted by about 15 days since the mid 20th century, while fall temperatures have shifted by about 10 days only. Uncertainty in 1947 climate normals is noticeably smaller than uncertainty in 2020 climate normals, consistent with the rapid warming near the end of the observed period, which makes the estimation more difficult.

Beyond this simple diagnosis, revised climate normals are an important tool to climate monitoring and to characterizing weather and climate events with respect to today's climate. We expect up-to-date climate normals to be of interest for such activities in the future.

### 3.4 Projections

We compute projections of annual and seasonal mean temperatures, constrained by both GSAT and regional temperature observations using the KCC method. The choice to consider both global and regional observations in implementing the constraint follows Qasmi and Ribes (2021), and is consistent with the choice discussed above for estimating past changes. It enables us to provide a consistent assessment of both past and future changes. As discussed in Section 2, only annual-mean observations are used in the constraint, so that summer and winter projections are constrained by the observed annual-mean warming.

Results are given for 4 SSP scenarios (Figure 5 and Table 1). As a general result, the combined observational constraint by GSAT and regional observations leads us to revise the CMIP6 projected warming upwards. For all seasons and scenarios, the CMIP6 best-estimate is revised upward, typically by 10% in the late 21st century. The observational constraint also leads to a significant narrowing of the 5-95% confidence range by 40% to 50% in the late 21st century, and even more in the near term. This affects primarily the lower bound of the confidence range, which is strongly revised upwards, and comes closer to the unconstrained CMIP6 multi-model mean. The upper bound is usually slightly shifted downwards – consistent with GSAT results. In the intermediate scenario SSP2-4.5, the expected annual mean warming in 2100 is assessed to be 3.8°C (2.9 to 4.8°C). Furthermore, as GHG emissions are still positive at that time in this scenario, temperature is still rising. Beyond the upward revision, the temporal shape of the temperature response is weakly affected by the constraint. Remarkably, in low (SSP1-2.6) or intermediate (SSP2-4.5) scenarios, the highest warming rate has occurred recently, or is occurring now (Figure S2). This suggests that the current period may be critical with respect to climate adaptation.

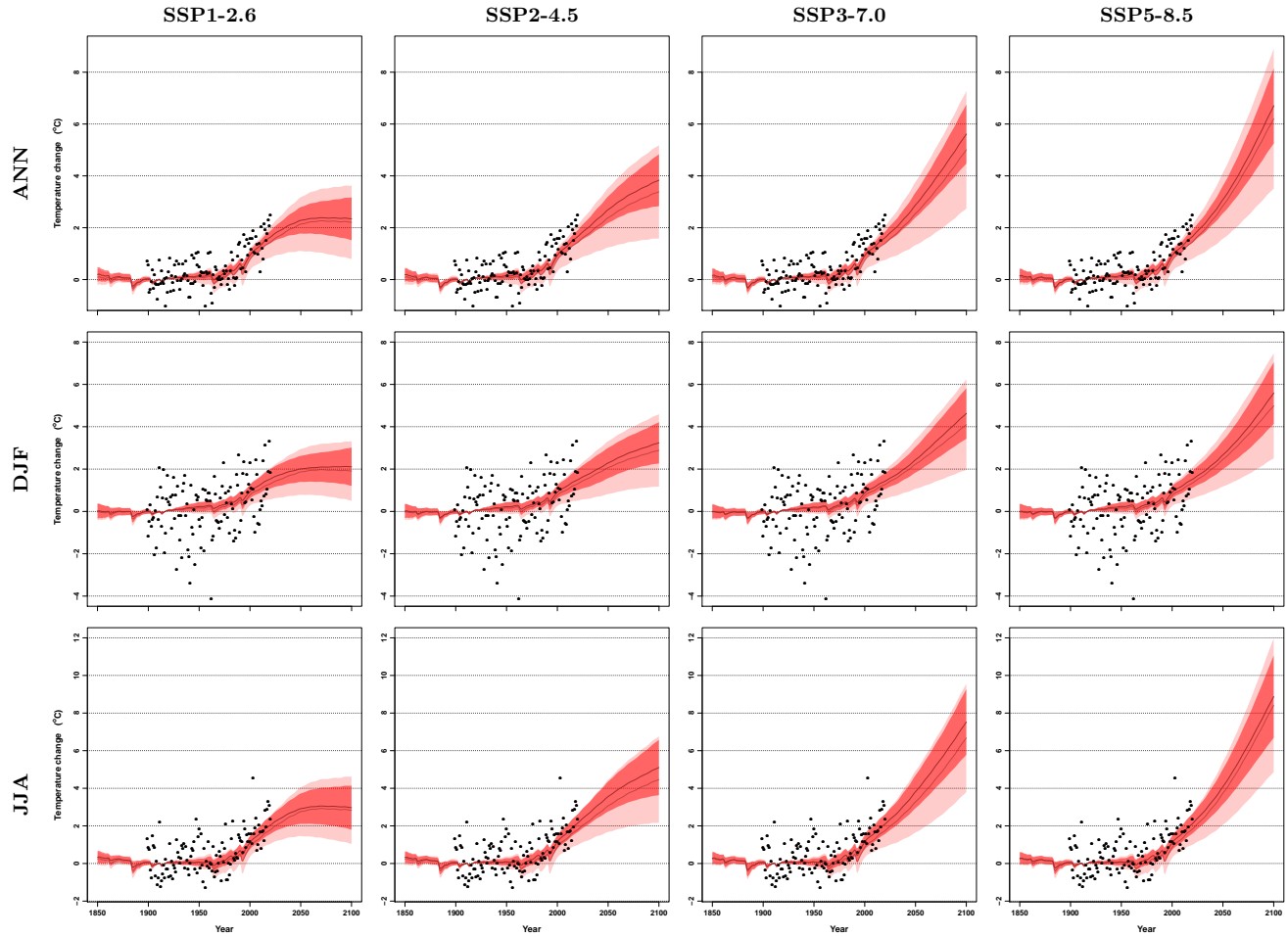

**Figure 5. Constrained mean temperature projections.** CMIP6 projections constrained by both global (GSAT) and regional observations for annual, winter and summer mean temperature over France, and for the 4 illustrative SSP scenarios considered in this study. Annual mean temperatures over France (black points) are compared to the unconstrained (pink) and constrained (red) 5 to 95% confidence ranges of the forced response, as estimated from CMIP6 models. All temperatures are anomalies with respect to the period 1900–1930.

**Table 1.** Changes in annual mean temperature over France, assessed from CMIP6 projections constrained by both global and regional observations, for selected time periods and the four illustrative emissions scenarios considered. Temperature differences are relative to the period 1900–1930 and are reported in °C.

| | Scenario | 2020–2040 Best est. | 2020–2040 5–95% | 2040–2060 Best est. | 2040–2060 5–95% | 2080–2100 Best est. | 2080–2100 5–95% | 2100 Best est. | 2100 5–95% |
|---|---|---|---|---|---|---|---|---|---|
| **ANN** | SSP1-2.6 | 1.9 | [ 1.6 to 2.2 ] | 2.3 | [ 1.8 to 2.8 ] | 2.4 | [ 1.6 to 3.1 ] | 2.3 | [ 1.5 to 3.1 ] |
| | SSP2-4.5 | 2 | [ 1.7 to 2.3 ] | 2.7 | [ 2.2 to 3.2 ] | 3.7 | [ 2.8 to 4.5 ] | 3.8 | [ 2.9 to 4.8 ] |
| | SSP3-7.0 | 2 | [ 1.7 to 2.3 ] | 2.9 | [ 2.4 to 3.4 ] | 5 | [ 4 to 6.1 ] | 5.6 | [ 4.5 to 6.8 ] |
| | SSP5-8.5 | 2.1 | [ 1.8 to 2.4 ] | 3.1 | [ 2.6 to 3.6 ] | 5.9 | [ 4.6 to 7.2 ] | 6.7 | [ 5.2 to 8.2 ] |
| **DJF** | SSP1-2.6 | 1.7 | [ 1.2 to 2.3 ] | 2 | [ 1.4 to 2.6 ] | 2.1 | [ 1.3 to 2.9 ] | 2.1 | [ 1.2 to 3 ] |
| | SSP2-4.5 | 1.8 | [ 1.2 to 2.3 ] | 2.3 | [ 1.6 to 3 ] | 3.1 | [ 2.2 to 3.9 ] | 3.2 | [ 2.3 to 4.2 ] |
| | SSP3-7.0 | 1.7 | [ 1.3 to 2.2 ] | 2.4 | [ 1.8 to 3 ] | 4.2 | [ 3.1 to 5.3 ] | 4.6 | [ 3.4 to 5.8 ] |
| | SSP5-8.5 | 1.9 | [ 1.4 to 2.4 ] | 2.7 | [ 2 to 3.4 ] | 4.9 | [ 3.6 to 6.3 ] | 5.6 | [ 4.1 to 7.1 ] |
| **JJA** | SSP1-2.6 | 2.4 | [ 1.8 to 3 ] | 2.9 | [ 2.1 to 3.7 ] | 3 | [ 2 to 4.1 ] | 3 | [ 1.8 to 4.1 ] |
| | SSP2-4.5 | 2.6 | [ 2.1 to 3.2 ] | 3.5 | [ 2.7 to 4.3 ] | 4.8 | [ 3.5 to 6.2 ] | 5.1 | [ 3.6 to 6.6 ] |
| | SSP3-7.0 | 2.6 | [ 2 to 3.2 ] | 3.8 | [ 2.9 to 4.7 ] | 6.7 | [ 5.2 to 8.3 ] | 7.5 | [ 5.8 to 9.3 ] |
| | SSP5-8.5 | 2.7 | [ 2.1 to 3.3 ] | 4.1 | [ 3.2 to 5 ] | 7.8 | [ 5.9 to 9.8 ] | 8.9 | [ 6.6 to 11.2 ] |

Seasonal mean results confirm the well-known enhanced summer warming over this region (Giorgi and Lionello, 2008; Terray and Boé, 2013; Lionello and Scarascia, 2018). The long-term summer to winter warming ratio (Table 1) is close to 1.5, and is not affected by the observational constraint (i.e., unconstrained projections exhibit the same ratio). This ratio is consistent with, although slightly higher than, the 1.3 ratio found in post-1947 observations. Expected mean temperature changes by 2100 in an intermediate emission scenario (SSP2-4.5) are assessed to be 3.2°C (2.3 to 4.2°C) in winter, and 5.1°C (3.6 to 6.6°C) in summer. Overall, winter and summer warming are expected to be about 15% lower than, and 30% higher than the annual mean warming, respectively, for all scenarios and time periods. These ratios are also consistent with recent observations (Figure 4).

A simple interpretation of the observational constraint results is as follows. The ensemble of CMIP6 models, from which our prior is derived, suggests that the late 21st century forced warming is tightly related to the forced warming in 2020. Although the KCC method is complex and uses the entire observed time-series to build the constraint, simply inferring the late 21st

century warming from the warming to date provides results consistent with the full method (Figure S3). This result suggests that the upward revision of the past forced warming directly results in an upward revision of the future forced warming. This near-linear relationship holds for all scenarios, and the warming ratios between future and past changes are very close at both global and regional scales. For the SSP2-4.5 scenario, the ratio of 2100 to 2020 forced warming is about 2.4 over France, and 2.5 globally (according to Ribes et al., 2021). This ratio increases to 4 in a very high emission scenario SSP5-8.5, both globally and regionally. The similarity between the global and regional ratios supports a pattern scaling hypothesis, and suggests that this ratio is directly driven by the increase in radiative forcing between 2020 and 2100. It also suggests that the fraction of warming offset by aerosols is fairly similar regionally and globally in 2020 (consistent with attribution results in subsection 3.2; the case would have been very different, e.g., in the 1980's). As a consequence of this near-linear relationship, a higher warming to date implies a higher 21st century warming. So, if our estimate of forced warming in 2020 were to be considered conservative (e.g., because recent observations point to higher levels of warming), then our estimate of future warming should also be considered conservative. Similarly, if GSAT observations were not used in the KCC constraint (i.e., the constraint uses regional observations only), then the projected 21st century warming ranges would be revised upward compared to those shown in this study. This underlines the importance of the detailed discussion on how to best estimate the forced warming to date (Figure 1), as this has direct implications on the assessed future warming.

The upward revision of projected warming values over France is a key result of this regional analysis. This result is somewhat unexpected, given the reported downward revision of projected GSAT changes using observational constraints (Lee et al., 2021), and the strong relationship between GSAT and regional changes (Figure 2). This key finding suggests that regional observations (i.e., not only global) now provide valuable information about on-going climate change, and that the observed record as a whole can be used to discard some of the putative forced responses simulated by climate models.

The recent IPCC AR6 stressed that there is a near-linear relationship between cumulative $CO_2$ emissions and global mean warming (Canadell et al., 2021). Since the warming over France also exhibits a near-linear relationship to global mean warming (Figure S4), this finding also applies to the regional scale warming. Thus, the expected future warming over France is expected to be near-linear on the cumulative $CO_2$ emissions (Figure S5). Non-GHG forcings like aerosols induce a slight deviation at low $CO_2$ emissions, but do not affect the near-linear relationship in the future, at least in the SSP scenarios considered here. This result implies that, both at the French and global levels, every tonne of $CO_2$ emission adds to the warming. Stabilizing the temperature at a given level therefore requires net-zero emissions, whatever the warming target is.

## 4    Discussion and conclusion

This study provides a revised assessment of past and future warming over France. As a key novelty, we combine available information from the latest generation of climate models and observed global and regional mean temperature records through the application of an observational constraint at the regional scale. This original technique revises model estimates of past and future warming upwards. This occurs despite the fact that GSAT observations tend to pull down our regional warming estimates.

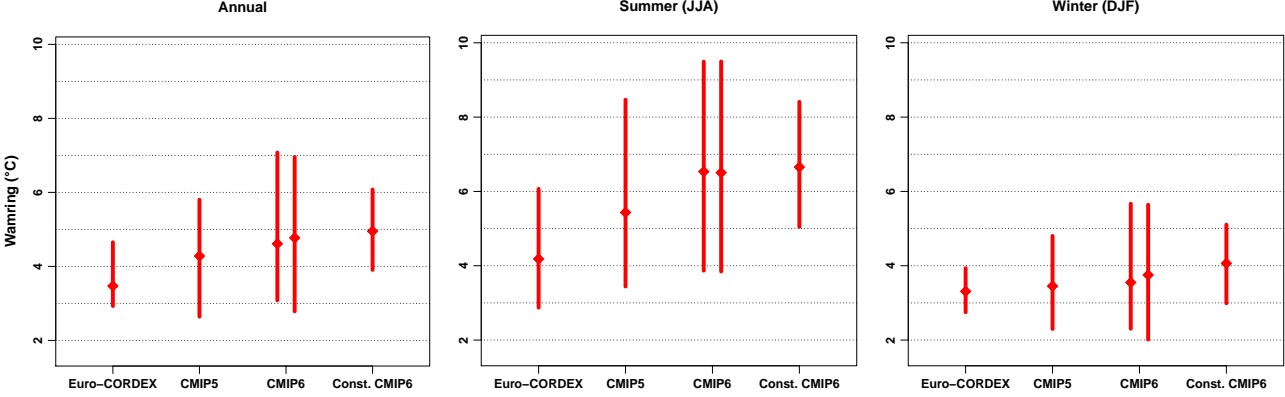

**Figure 6. Forced warming estimates from various multi-model ensembles.** The constrained CMIP6 ranges of mean temperature change are compared to those from the Euro-Cordex, CMIP5 and CMIP6 ensembles (all unconstrained). The comparison is made for the 2070–2098 vs 1971–2000 warming (periods that are covered by all model experiments), in the RCP8.5 (Euro-Cordex and CMIP5) or SSP5-8.5 (CMIP6) scenarios. All confidence ranges are 5-95% ranges, with the median used as a central estimate. The two CMIP6 ranges are derived with/without assuming a Gaussian distribution: on the left, quantiles are directly estimated from the sample of CMIP6 models (consistent with Euro-Cordex and CMIP5); on the right, a Gaussian distribution is assumed for that sample (consistent with the prior used in the observational constraint).

Specifically, regional observations drive the estimates up more than global observations drive them down – consistent with a strong upward revision of the regional to global warming ratio compared to the raw model results.

Combining these two lines of evidence, we assess the forced warming in 2020 wrt 1900-1930 to be 1.66 [1.41 to 1.90] °C, which lies in the upper range of the unconstrained CMIP6 estimates. Human-induced warming over the same period is estimated to be 1.63 [1.39 to 1.88] °C, implying that France observed warming is almost entirely human-induced. The current rate of warming is found to be 0.36 [0.27 to 0.45] °C/decade, to which aerosol recovery contributes significantly. Projected warming in response to an intermediate SSP2-4.5 emission scenario is assessed to be 3.8°C (2.9 to 4.8°C) in 2100, and rises up to 6.7 [5.2 to 8.2] °C in a very high emissions SSP5-8.5 scenario. Still in the SSP2-4.5 scenario, seasonal warming is estimated to be 3.2°C (2.3 to 4.2°C) in winter and 5.1°C (3.6 to 6.6°C) in summer.

## 4.1 Comparison to other multi-model ensembles

Comparing our results with those based on previous generations of climate model ensembles (Figure 6) reveals that our assessed ranges lie substantially higher than previously reported. Reasons explaining why our constrained range lies in the upper range of the unconstrained CMIP6 have been discussed already and are related to taking into account regional observations.

The CMIP5 ensemble exhibits a lower warming than CMIP6 (about 10% lower on the annual mean) and a slightly lower spread. These discrepancies are consistent with differences found at the global scale (i.e., higher and more spread out GSAT

changes and climate sensitivity, e.g., Forster et al., 2019). Subtle changes in scenarios, from Representative Concentration Pathways (RCPs) in CMIP5 to Shared Socio-economic Pathways (SSPs) in CMIP6 (although the nominal level radiative forcing in 2100 is the same in the two generations), have been shown to modestly strengthen the late 21st century warming in CMIP6 (Fyfe et al., 2021). But the upward shift in climate sensitivity (e.g., Transient Climate Response, TCR, or Equilibrium Climate Sensitivity, ECS) is responsible for most of the difference between CMIP5 and CMIP6. Our case study suggests that the well documented large spread in climate sensitivity among CMIP6 models, and in particular the presence of high-sensitivity models, might be useful to cover a larger spectrum of regional responses to an increased greenhouse effect. Specifically, our constrained CMIP6 range sometimes exceeds the upper bound of the CMIP5 range. In other words, high-sensitivity CMIP6 models are useful to sample the upper-end of the regional response, although evidence suggests that the high regional warming is related to a high-end regional warming ratio rather than a high-end global climate sensitivity.

The EURO-CORDEX ensemble is an ensemble of high-resolution (∼12km) area-limited Regional Climate Models (RCMs), driven by a limited subset of CMIP5 global models. It exhibits less warming (especially in the annual and summer temperature), and less spread than the whole CMIP5 ensemble (Figure 6). This discrepancy is already described in the literature (e.g., Boé et al., 2020a), and various explanations have been proposed or shown to contribute to the reduced warming. The absence of time-varying anthropogenic aerosols (Boé et al., 2020a; Gutiérrez et al., 2020) and $CO_2$ physiological effect (Schwingshackl et al., 2019; Boé, 2021) in most EURO-CORDEX RCMs has been suggested to be responsible for a large part of the differences in summer warming between EURO-CORDEX RCMs and CMIP5 models – with therefore more realistic warming expected in CMIP5 models. Conversely, other studies suggested that the RCMs should be considered as more reliable, either owing to their higher spatial resolution (leading to improved physical processes, e.g., Bartók et al., 2017), or their reduced climatological biases (e.g., Sørland et al., 2018). In this ongoing debate, our results provide a new line of evidence, based on observations, that the summer warming projected by EURO-CORDEX RCMs is unrealistically small, and less realistic than that from their forcing GCMs.

Considering the 4 multi-model ensembles, it appears that our constrained CMIP6 range points to higher values than all previous ranges. The discrepancy with EURO-CORDEX is particularly large and potentially problematic in an adaptation planning perspective. Indeed, for the annual and summer temperature projections, the EURO-CORDEX best estimate lies outside the constrained CMIP6 range, and vice-versa (the constrained CMIP6 best-estimate lies outside of the EURO-CORDEX range). As the CMIP6 constrained range is tightly related to observed changes over the last 120 yr, this suggests that some of the EURO-CORDEX models might not be able to simulate a past warming consistent with observations. Testing this hypothesis is currently impossible, given the limited length of EURO-CORDEX runs and the lack of single model ensemble members to sample internal variability appropriately. Further investigation will be needed to better understand physical reasons behind this discrepancy.

## 4.2 Extension of these results

The results of our study only relate to Mainland France. Replication of these results to other areas or countries may be of interest. With regard to Western Europe specifically, since the observations made in France can be considered representative of

a wider region, it is expected that some of our results will generalise beyond the borders of France, at least qualitatively – in particular, the fact that the constraint based on regional observations draws the expected warming upwards.

The approach and methods proposed in this article could be used on a regular basis, e.g., to monitor and update the assessed forced warming ranges annually. Warming to date is a key indicator, e.g., to check the crossing of selected thresholds. Beyond past warming, it is now also possible to take advantage of the latest observations to further refine estimates of expected warming in response to various emission scenarios – even if the addition of each individual year will have a limited effect on the late 21st-century estimates.

Extending the diagnoses to other variables is also an important issue. Beyond mean temperature, Terray and Boé (2013) provided important diagnoses on the expected changes in mean precipitation. Due to the much smaller signal-to-noise ratio for precipitation than for temperature (especially over France where precipitation trends remain partly uncertain), the application of observational constraints does not yet allow to refine precipitation projections. However, describing the expected changes in precipitation based on the unconstrained CMIP6 ensemble is of interest, and is done in Supplementary Material. In summary, we find a winter wetting (+4% to +35% in 2070-2098 wrt 1971-2000) and a summer drying (-14% to -52%), while changes in the annual rainfall are more limited (-11% to +7%). These results are consistent with previous multi-model ensembles (Euro-Cordex, CMIP5), although seasonal changes are slightly more pronounced in CMIP6, in line with the projected enhanced precipitation seasonality highlighted over Europe (Douville et al., 2021). Although post-1960 observations do not exhibit long-term trends (in line with model results), early 20th century observations exhibit a wetting winter trend that seems inconsistent with model results. The interpretation of these differences, and the possible use of observational constraints for precipitation, will be the subject of future research.

This work also raises some new questions.

The assessed regional warming rate appears to be particularly high over France, if compared to CMIP6 models. New questions arise from this finding. First, could observational issues (e.g., urban heat islands effect, the homogenization procedure, etc) contribute to the very high warming trend over the recent period? Second, is there any reason why CMIP-style models would systematically underestimate the regional warming rate (e.g., due to their construction or resolution)? If so, using them as a "priori" could be questioned, and/or they could be corrected (i.e., unbiased). Third, what are the physical processes responsible for this difference, or likely to explain a high regional warming rate? Changes in the large-scale atmospheric circulation could play a role (Boé et al., 2020b) as well as other factors.

Another question concerns the residual difference between observations over the last 10 or 20 years and our assessed forced response (the latter remains substantially below the measured mean temperature of the last decade). Our work suggests that this difference is statistically consistent with internal variability. But internal variability over this period could be characterized or even assessed, e.g., using information on atmospheric dynamics (via analogs, weather regimes), or teleconnections. Accounting for such information, can internal variability really explain the difference between measurements and our assessed forced response? More generally, would it be possible to make the observational constraints even more accurate by taking into account the available information about observed internal variability (i.e., a partial denoising of the observations)?

### 4.3 Implication for modeling activities and climate services

The emergence of observational constraints such as those presented in this paper raises a number of questions about the development of future climate models (either global or regional). First, as outlined above, assessing the agreement or disagreement of a particular model with the available observations requires single-model ensemble members covering the whole observed period – which has direct implications on the resolution or numerical cost of the model in question. Second, an important challenge for climate services is to provide a reduced number of simulations to sample the uncertainty in the magnitude of future warming – this deserves some discussion.

Various studies have proposed, through weighting methods, to select models that are consistent with recent observations (Brunner et al., 2020; Liang et al., 2020) – and therefore observational constraints. This approach has proven to be effective for global mean temperature. However, due to the small number of CMIP models available, it becomes limited very quickly if several features have to be assessed simultaneously. In this study where we consider only 2 variables (global and regional warming), only one model (ACCESS-CM2, Figure 2) manages to satisfy both global and regional constraints. Adding one or more other constraints would quickly lead to a situation where none of the available models satisfies all constraints at once. Consequently, the construction of climate models capable of satisfying different observational constraints remains a challenge, and will certainly require new uncertainty sampling and/or calibration techniques in the future.

Focusing on the future regional warming alone, we find that a subset of CMIP6 models can approximately sample the range of values retained by the observational constraint (despite the fact that many CMIP6 models fall outside of this range). Importantly, we notice that some high-sensitivity models are useful to appropriately sample that range. This result suggests that excluding systematically high-sensitivity CMIP6 models, based on GSAT considerations only, might not be the best practice for regional scale studies.

Unlike the CMIP6 ensemble, the EURO-CORDEX ensemble of high-resolution simulations does not cover the range of values retained by the observational constraint. The next EURO-CORDEX ensemble, which will be driven by CMIP6, may cover this interval better. But in any case, it seems relevant to look for alternatives in order to provide representative realisations. These alternatives could include statistical downscaling of CMIP6 simulations, the realisation of regional simulations using nudging techniques, or considering regional simulations at a "given regional temperature level" – similar to the use of "Global Warming Level" in the IPCC AR6.

*Code and data availability.* Code and data to reproduce the key figures of this study are available at `https://gitlab.com/ribesaurelien/france_study`, and as a Zenodo archive `https://doi.org/10.5281/zenodo.6029160` (Ribes, 2022).

## Appendix A: Climate models used

### A1  List of CMIP6 models used

ACCESS-CM2, ACCESS-ESM1-5, AWI-CM-1-1-MR, CAMS-CSM1-0, CanESM5-CanOE, CanESM5, CESM2, CESM2-WACCM, CMCC-CM2-SR5, CNRM-CM6-1-HR, CNRM-CM6-1, CNRM-ESM2-1, EC-Earth3-Veg, FGOALS-f3-L, FGOALS-g3, GISS-E2-1-G, INM-CM4-8, IPSL-CM6A-LR, MIROC6, MIROC-ES2L, MPI-ESM1-2-HR, MPI-ESM1-2-LR, MRI-ESM2-0, NorESM2-LM, NorESM2-MM, TaiESM1, UKESM1-0-LL (27 CMIP6 models).

### A2  List of CMIP5 models used

ACCESS1-0, ACCESS1-3, bcc-csm1-1-m, bcc-csm1-1, BNU-ESM, CanESM2, CCSM4, CESM1-BGC, CESM1-CAM5, CESM1-WACCM, CMCC-CESM, CMCC-CM, CNRM-CM5, CSIRO-Mk3-6-0, EC-EARTH, FGOALS-g2, FIO-ESM, GFDL-CM3, GFDL-ESM2G, GFDL-ESM2M, GISS-E2-H, GISS-E2-R, HadGEM2-AO, HadGEM2-CC, HadGEM2-ES, inmcm4, IPSL-CM5A-LR, IPSL-CM5A-MR, IPSL-CM5B-LR, MIROC5, MIROC-ESM, MIROC-ESM-CHEM, MPI-ESM-LR, MPI-ESM-MR, MRI-CGCM3, NorESM1-M, NorESM1-ME (37 CMIP5 models).

### A3  List of EURO-CORDEX models used

See Table A1.

| Euro-CORDEX RCM | Driving CMIP5 GCM |
| --- | --- |
| CLMcom-CCLM4-8-17 | CNRM-CERFACS-CNRM-CM5 |
| CLMcom-ETH-COSMO-crCLIM-v1-1 | CNRM-CERFACS-CNRM-CM5 |
| CNRM-ALADIN53 | CNRM-CERFACS-CNRM-CM5 |
| CNRM-ALADIN63 | CNRM-CERFACS-CNRM-CM5 |
| DMI-HIRHAM5 | CNRM-CERFACS-CNRM-CM5 |
| GERICS-REMO2015 | CNRM-CERFACS-CNRM-CM5 |
| KNMI-RACMO22E | CNRM-CERFACS-CNRM-CM5 |
| MOHC-HadREM3-GA7-05 | CNRM-CERFACS-CNRM-CM5 |
| RMIB-UGent-ALARO-0 | CNRM-CERFACS-CNRM-CM5 |
| SMHI-RCA4 | CNRM-CERFACS-CNRM-CM5 |
| CLMcom-CCLM4-8-17 | ICHEC-EC-EARTH |
| CLMcom-ETH-COSMO-crCLIM-v1-1 | ICHEC-EC-EARTH |
| ICTP-RegCM4-6 | ICHEC-EC-EARTH |
| KNMI-RACMO22E | ICHEC-EC-EARTH |
| MOHC-HadREM3-GA7-05 | ICHEC-EC-EARTH |
| SMHI-RCA4 | ICHEC-EC-EARTH |

| | |
|---|---|
| DMI-HIRHAM5 | ICHEC-EC-EARTH |
| DMI-HIRHAM5 | IPSL-IPSL-CM5A-MR |
| GERICS-REMO2015 | IPSL-IPSL-CM5A-MR |
| IPSL-INERIS-WRF331F | IPSL-IPSL-CM5A-MR |
| KNMI-RACMO22E | IPSL-IPSL-CM5A-MR |
| SMHI-RCA4 | IPSL-IPSL-CM5A-MR |
| CLMcom-CCLM4-8-17 | MOHC-HadGEM2-ES |
| CLMcom-ETH-COSMO-crCLIM-v1-1 | MOHC-HadGEM2-ES |
| CNRM-ALADIN63 | MOHC-HadGEM2-ES |
| DMI-HIRHAM5 | MOHC-HadGEM2-ES |
| ICTP-RegCM4-6 | MOHC-HadGEM2-ES |
| KNMI-RACMO22E | MOHC-HadGEM2-ES |
| MOHC-HadREM3-GA7-05 | MOHC-HadGEM2-ES |
| SMHI-RCA4 | MOHC-HadGEM2-ES |
| CLMcom-CCLM4-8-17 | MPI-M-MPI-ESM-LR |
| CLMcom-ETH-COSMO-crCLIM-v1-1 | MPI-M-MPI-ESM-LR |
| CNRM-ALADIN63 | MPI-M-MPI-ESM-LR |
| DMI-HIRHAM5 | MPI-M-MPI-ESM-LR |
| ICTP-RegCM4-6 | MPI-M-MPI-ESM-LR |
| KNMI-RACMO22E | MPI-M-MPI-ESM-LR |
| MOHC-HadREM3-GA7-05 | MPI-M-MPI-ESM-LR |
| MPI-CSC-REMO2009 | MPI-M-MPI-ESM-LR |
| SMHI-RCA4 | MPI-M-MPI-ESM-LR |
| UHOH-WRF361H | MPI-M-MPI-ESM-LR |
| GERICS-REMO2015 | MPI-M-MPI-ESM-LR |
| CLMcom-ETH-COSMO-crCLIM-v1-1 | NCC-NorESM1-M |
| CNRM-ALADIN63 | NCC-NorESM1-M |
| DMI-HIRHAM5 | NCC-NorESM1-M |
| GERICS-REMO2015 | NCC-NorESM1-M |
| ICTP-RegCM4-6 | NCC-NorESM1-M |
| KNMI-RACMO22E | NCC-NorESM1-M |
| MOHC-HadREM3-GA7-05 | NCC-NorESM1-M |
| SMHI-RCA4 | NCC-NorESM1-M |

Table A1: List of EURO-CORDEX models used, with their driving GCM (49 RCM/GCM pairs in total).

## Appendix B: Details on the observational constraint method

### B1  Observational operator $H$

The observational operator $H$ is very simple in this study. Observations in $\mathbf{y}$, either at the global scale ($T_{\text{glo}}^{\text{obs}}$) or at the regional scale ($T_{\text{reg}}^{\text{obs}}$), are representative of the global ($T_{\text{glo}}^{\text{all}}$) or regional ($T_{\text{glo}}^{\text{obs}}$) forced responses in $\mathbf{x}$, respectively, plus some noise related to internal climate variability. As the full 1850–2100 period is not covered by observations, $H$ also extracts the appropriate years from $\mathbf{x}$.

As a result, $H$ can be written as a block matrix

$$H = \begin{pmatrix} I_{1850:2020} & 0 & 0 & 0 & 0 \\ 0 & I_{1899:2020} & 0 & 0 & 0 \end{pmatrix}, \tag{B1}$$

where $I_{y_1:y_2}$ is similar to an identity matrix, but with $0$ on the diagonal outside the period from year $y_1$ to year $y_2$ – so it extracts years from $y_1$ to $y_2$.

### B2  Estimation of $\Sigma_{\mathbf{x}}$

$\Sigma_{\mathbf{x}}$ basically describes the spread in the forced responses of CMIP6 models. $\Sigma_{\mathbf{x}}$ is derived in two steps. First, for each CMIP6
model considered, we estimate the forced response in each of the $T$ vectors shown in Eq (2). This includes the forced response in GSAT, in annual and seasonal mean temperature over France, and the response to specific forcings (i.e., NAT-only or GHG-only). As a result, we have a sample of 27 model estimates of $\mathbf{x}$ – one for each CMIP6 model considered. Second, we estimate $\Sigma_{\mathbf{x}}$ as the sample covariance matrix over this sample of 27 model estimates.

Using a sample covariance estimate has the side effect of producing a highly degenerated estimate for $\Sigma_{\mathbf{x}}$: while $\Sigma_{\mathbf{x}}$ is a
540 $n_{\mathbf{x}} \times n_{\mathbf{x}}$ matrix ($n_{\mathbf{x}} = 1426$), the rank of our estimate is equal to 26 (since 27 CMIP6 models are being considered). While this choice could be debated, the KCC method can be run in this way, as $\Sigma_{\mathbf{x}}$ does not need being inverted to derive the parameters $\mu_p$ and $\Sigma_p$ of the posterior distribution. The only matrix that has to be inverted in this calculation is $S = (H\Sigma_{\mathbf{x}}H' + \Sigma_{\mathbf{y}})$. In our case, $S$ is guaranteed of being invertible thanks to $\Sigma_{\mathbf{y}}$.

### B3  Estimation of $\Sigma_{\mathbf{y}}$

According to Eq (3), $\mathbf{y} = \left( \mathbf{T}_{\text{glo}}^{\text{obs}}, \mathbf{T}_{\text{reg}}^{\text{obs}} \right)$, where the length of $\mathbf{T}_{\text{glo}}^{\text{obs}}$ and $\mathbf{T}_{\text{reg}}^{\text{obs}}$ are $n_{\mathbf{y}}^{\text{glo}} = 171$ and $n_{\mathbf{y}}^{\text{reg}} = 122$, respectively. As a consequence, $\Sigma_{\mathbf{y}}$ can be decomposed as

$$\Sigma_{\mathbf{y}} = \left( \begin{array}{c|c} \Sigma_{\mathbf{y}}^{\text{glo}} & 0 \\ \hline 0 & \Sigma_{\mathbf{y}}^{\text{reg}} \end{array} \right), \tag{B2}$$

where $\Sigma_{\mathbf{y}}^{\text{glo}}$ and $\Sigma_{\mathbf{y}}^{\text{reg}}$ are square matrices of dimension $n_{\mathbf{y}}^{\text{glo}} \times n_{\mathbf{y}}^{\text{glo}}$ and $n_{\mathbf{y}}^{\text{reg}} \times n_{\mathbf{y}}^{\text{reg}}$, respectively. In Eq (B2), off-diagonal blocks are assumed to be $0$, corresponding to independent errors at the global and regional levels – this is further discussed below.

In KCC, the observational error corresponds to the difference between observations and the forced response. So, measurement uncertainty and internal variability contribute to $\Sigma_{\mathbf{y}}$. Estimating $\Sigma_{\mathbf{y}}$ requires statistical modelling of these two terms. Regarding the GSAT block, $\Sigma_{\mathbf{y}}^{\mathrm{glo}}$, we use the same estimate as Ribes et al. (2021). Internal variability is estimated as a sum of two Auto-Regressive processes of order 1 (AR1). Measurement uncertainty is estimated from the HadCRUT5 ensemble. Regarding the regional block, $\Sigma_{\mathbf{y}}^{\mathrm{reg}}$, we follow Ribes et al. (2016) and assume that internal variability in annual mean temperature

over France follows an AR1($\alpha$=0.2) process. We assume no measurement error in regional temperature observations, so only internal variability contributes to $\Sigma_{\mathbf{y}}^{\mathrm{reg}}$. Neglecting measurement uncertainty is acceptable here since the uncertainty related to internal variability alone is quite large at the regional scale ($\sigma$=0.51°C for each single year). The variance of both global and regional internal variability is derived from observations, after subtracting the CMIP6 multimodel mean (crude) estimate of the forced response. Lastly, we assume global and regional internal variability to be independent, as observed data exhibit no

significant correlation between these two spatial scales. The obtained estimate of $\Sigma_{\mathbf{y}}$ is invertible, as the two blocks $\Sigma_{\mathbf{y}}^{\mathrm{glo}}$ and $\Sigma_{\mathbf{y}}^{\mathrm{reg}}$ are invertible themselves.

*Author contributions.*   A.R. designed the study, run the calculation and produced most figures. J.B. performed a large part of data processing, and produced some figures. Q.S. contributed to the main code and re-run most analyses independently. B.D. provided observational data, and guidance on how to use it. All authors participated in the interpretation of the results and the writing of the manuscript.

*Competing interests.*   The authors declare no competing interest.

*Acknowledgements.*   We thank the climate modeling groups involved in CMIP6, CMIP5 and the EURO-CORDEX exercises for producing and making available their simulations. A.R., Q.S, H.D. acknowledge support by Météo France, CNRS, the European Union's Horizon 2020 Research and Innovation Program under the EUCP (grant agreement 776613) and the CONSTRAIN (grant agreement 820829) projects.

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
