# Peer review of "An updated assessment of past and future warming over France based on a regional observational constraint"

_Earth System Dynamics, 2022_

## Referee Comment (RC3)

Review - *An updated assessment of past and future warming over France based on a regional observational constraint* by Ribes et al (https://doi.org/10.5194/esd-2022-7)

This paper reports on the application of a Bayesian technique that provides estimates of the distributions of past and future regional warming conditional on observed changes in regional and global mean temperatures. Conditioning on the observed changes produces historical warming estimates for France that are lower than recorded in the observations, but higher than obtained by calculating the CMIP6 multi-model mean. It also produces projections of future change that tend to warm more than the CMIP6 multi-model mean and have lower uncertainty than the regional projections from the CMIP6 models. The paper also considers how the constrained warming estimates can be used to provide estimates of updated 30-year climatologies for moving 30-year windows that account for the evolving climate. Precipitation is very briefly considered in the supplementary information.

While the paper is very nicely written, it feels as if it has both too much material, and in some senses, too little material.

First, there is very little discussion of the assumption that the "models are indistinguishable from the truth", but this is at very core of the technique. This is important to discuss, because it implies both model democracy and model equity with observations. This seems counter to approaches that would identify constraints based on model performance relative to observations and concerns that the very high climate sensitivity models found in CMIP6 may not be realistic (e.g., Hausfather et al, 2022, https://www.nature.com/articles/d41586-022-01192-2).

Second, the authors rely on previous papers that they have published (Ribes et al, 2021) or submitted (Qasmi and Ribes, 2021) to present the methodology. While I don't have strong objections to this, it does mean that this paper is not as self-contained as it might be. This creates some challenges for readers who are interested in the methodological details since some of the details of the application of the method are different in this paper than in Ribes et al (2021) and Qasmi and Ribes (2021).

Third, the paper does not provide very much information about the observed regional temperature time series and how it is computed. A hard to obtain paper by Mestre et al (2013) is cited describing the homogenization software that is used in the production of the dataset, but apparently homogenizations is only applied to the early part of the station records that go into the French national thermal index – data from 1947 onwards are not homogenized. When I went searching for the French « Indice Thermique Nationale », I found it surprisingly difficult to find a description. I did, ultimately, find a map showing the locations of the 30 long-running stations that comprise this temperature indicator, and wasn't surprised to see that many of those stations appear to be located in or near urban centers. A potential concern about the data, therefore, is whether these stations are affected by the gradual expansion or intensification of urban heat islands, in which case one might worry that the recorded warming may not be representative of the warming experienced by Metropolitan (mainland) France as a

whole. The paper hints that declining aerosol burdens may have something to do with the rapid warming in the region, but it may also be the case that the data are not as homogeneous as thought.

Finally, I think it should be recognized that the technique apparently does not account for uncertainty in covariance and mean vector estimates that enter into calculation of the posterior distribution for regional mean temperatures. To do so, wouldn't it be necessary to use a more complex hierarchical Bayes model that uses priors on these parameters to describe their potential uncertainty?

Some specific comments:

72-73: This sounds like it might be unnecessarily complex. Since the interest is just in the Metropolitan France spatial mean temperature, does this interpolation scheme simply amount to weighting each climate model grid box by the fraction of the box that lies within Metropolitan France? Is there perhaps a step here that is not described, such as using only the 30 10-km grid boxes that contain the 30 stations that contribute to the ITh to calculate the model version of the ITh?

82: It does seem a bit ironic that, on the one hand, the authors repeat concerns about pattern scaling (lines 42-47), but then rely on it quite heavily throughout the rest of the paper (note that I personally would do so as well).

99: Is the fact that the 30 stations are all at low elevation a limitation?

120-126: I was initially misled by the structure of equation (1) and the explanation of the matrix H. It some time for me to realize that H's function was simply to select a piece of the vector x (i.e., it consists of zeros, except for one or more embedded identity sub-matrices that live within H). Describing the methodology more completely within the paper so that one doesn't have to go searching in related papers would greatly help the reader. This could be done in the supplement, for example, which would probably be better used for that purpose than for dealing quickly with precipitation in any case) .

154: I'm not sure that "measurement uncertainty" would be quite the right description of the information that is represented by the spread of the HadCRUT5 ensemble. Instrumental measure error might be part of that spread, but uncertainty associated with incomplete instrumental coverage and missing data, the adjustment of different types of SST data, the calculation of local anomalies, etc, all contribute, do they not? What we learn, I think, is something about the uncertainty of the estimated global mean temperature anomaly for each year.

158-160: In this case, I think it is appropriate to talk about measurement error, and to assume that it is small relative to internal variability (provided that the stations have very little if any missing data).

242-255: This seems to be about attribution, and yet the next subsection has the title "Attribution". Section 3.2 could perhaps be better entitled "Contributions to warming from different forcing agents" or some such.

295: The discussion of the climate normals adds an interesting twist to the paper, but before making a lot of the apparent lack of uniformity of warming and discussing evidence for a change of seasonality, shouldn't one add uncertainty bands to Figure 4?

332: It would be good to add a reference to a paper that describes and analyzes this "well-known" enhancement.

464-469: Should one also ask a question about the observations? Using a classical D&A technique, Sun et al. (2016, doi: 10.1038/NCLIMATE2956), showed that about 1/3rd of the recorded warming in China between 1951 and 2010 could be attributed to urban warming, and that the remaining warming of about 1°C was much more consistent with the global land mean warming over that period.

---

## Author Response (AR1)

**Response to Review Comment #1**

*Using the newly developed statistical method (Kriging for Climate Change; KCC) and observational records, this paper aims to constrain the uncertainty of IPCC model simulated past and projected future warming under various scenarios over mainland France. Authors found that anthropogenic influences dominates the warming in 2020, and projects a significant ~3.8°C and ~6.7°C warming in the end of 21 century under CMIP6 SSP245 and SSP585 scenario, respectively. This paper also compares CMIP6 with CMIP5 and regional EURO-CORDEX simulations, and a thorough discussion has been made. I'm glad to say that I haven't got much to do as a reviewer of this excellent manuscript. It is well-written, clear, thorough, and of great scientific interest. I recommend it for publication at current form.*

We are grateful to Anonymous Referee #1 for this very positive comment about our manuscript.

**Response to Review Comment #2**

*Title: An updated assessment of past and future warming over France based on a regional observational constraint*
*Authors: Ribes et al.*

*Summary:*
*This paper assesses the past and future warming over France at the regional scale. One highlight of this paper is about the usage of Kriging for climate change, a method based on Bayesian Statistics, to get the posterior estimation of the projections after "assimilating" observations, which should substantially reduce the estimation uncertainties. As a researcher working on data assimilation, it is very inspiring and enlightening to see how data assimilation methods can be used for climate projections. The paper is well-written and clear. It would be great if the authors can show more details about the Kriging for climate change (KCC).*

*Recommendation: Minor revision*

We are grateful to Reviewer 2 for this general positive comment.

*Major Comments:*

*More detailed procedure describing the KCC is needed (which can be put in the appendix). Especially how you set the prior covariance for x in eq.(2). You mentioned at Line 150 that "$\mu_x$ and $\Sigma_x$ are estimated as the sample mean and covariance of the CMPI6 model forced responses." But how do you calculate $\Sigma_x$ exactly? How does $\Sigma_x$ look like? For data assimilation, the setting of prior error covariance requires a lot of efforts. What's the dimension of $\Sigma_x$. Is it diagonal or block diagonal? Does Kriging requires the calculation of inverse of $\Sigma_x$?*

We have revised the Section 2.3 about "Statistical methods", in order to add some specific details about the methodology, and we have added an Appendix B about "Details on the observational constraint method". Within Appendix B, one full paragraph is devoted to the estimation of $\Sigma_x$. In this paragraph, we make it clear that $\Sigma_x$ is a non-diagonal, degenerated matrix, which does not need being inverted (it is actually not invertible).
The dimension of x and $\Sigma_x$ is mentioned in both main text and Appendix B.
Note that revision of the Method section was also asked by Reviewer 3, and we considered all comments simultaneously while revising that section.

*You mentioned near line 130 that "x…where each element is an entire 1850-2100 time series of the forced response.", but what is the exact dimension of x? If x is large, how to you invert $\Sigma_x$?*

The dimension of x is $n_x=1426$, which is now mentioned explicitly in both the main text and the new Appendix B, with explanation l.132-136. As a result, $\Sigma_x$ is a large matrix, but it does not need being inverted.

*Near Line 120: what's the exact dimension of your vector y?*

The dimension of y is n_y=293. This has been added l.139.

*Near line 160, you mentioned that no measurement error is assumed. Do you mean Sigma_y = 0? Can you give an explanation what's the impact of setting Sigma_y = 0 in KCC, specially how does your influence influence the posterior?*

The estimation of Sigma_y is now detailed in Appendix B. We make it clear that measurement errors are only neglected at the regional scale. Assuming no measurement uncertainty does not mean that Sigma_y=0, since internal variability is another (strong) contributor to Sigma_y. We have added some discussion about that specific assumption (no measurement uncertainty at the regional scale). The revised presentation of the methodology also stresses that the method could not work if Sigma_y=0, as the matrix S in l.541 would no longer be invertible in such a case.

**Response to Review Comment #3 (Francis Zwiers)**

*Review - An updated assessment of past and future warming over France based on a regional observational constraint by Ribes et al (https://doi.org/10.5194/esd-2022-7)*

*This paper reports on the application of a Bayesian technique that provides estimates of the distributions of past and future regional warming conditional on observed changes in regional and global mean temperatures. Conditioning on the observed changes produces historical warming estimates for France that are lower than recorded in the observations, but higher than obtained by calculating the CMIP6 multi-model mean. It also produces projections of future change that tend to warm more than the CMIP6 multi-model mean and have lower uncertainty than the regional projections from the CMIP6 models. The paper also considers how the constrained warming estimates can be used to provide estimates of updated 30-year climatologies for moving 30-year windows that account for the evolving climate. Precipitation is very briefly considered in the supplementary information.*

*While the paper is very nicely written, it feels as if it has both too much material, and in some senses, too little material.*

*First, there is very little discussion of the assumption that the "models are indistinguishable from the truth", but this is at very core of the technique. This is important to discuss, because it implies both model democracy and model equity with observations. This seems counter to approaches that would identify constraints based on model performance relative to observations and concerns that the very high climate sensitivity models found in CMIP6 may not be realistic (e.g., Hausfather et al, 2022, https://www.nature.com/articles/d41586-022-01192-2).*

This is a very important comment. There are two key reasons why we think our approach is valid, even if some models have been spotted as being too sensitive, eg, in IPCC AR6.

First, evidence that models are too sensitive comes primarily from historical GSAT observations. We account for those observations in our method, as they are part of y. So, our constraint does account for GSAT observational constraint. The results from Ribes et al. (2021) even suggest that our GSAT-only constraint leads to ranges very consistent with IPCC AR6 (see Fig. 4.11), and Hausfather et al – so the agreement is quantitative too. If the prior was modified to exclude "hot models", then there would be a risk for circular reasoning: information from y would be used to define the prior pi(x), which is definitely not a good practice in Bayesian statistics. Instead, we seek for a prior representative of the knowledge in the absence of any consideration of the historical GSAT observations. We think that model democracy remains a reasonable choice in this context – and to some extent, this is supported by Ribes et al (2021) results.

Second, our results suggest there is an a posteriori argument for keeping those models. Our Figure 2 (looking at values on the y-axis only) suggests that in terms of recent climate, the Hausfather's "hot models" do a better job at the regional scale (ie, they are consistent with observations, where a majority of other models are not). In our view, this provides some evidence that "hot models" can be more consistent with observations over a specific region.

This happens despite the fact that "hot models" seem inconsistent with GSAT observations and should be excluded from GSAT projections. In other words, GSAT might not be the only variable / criteria for selecting / weighting models, and a selection based on GSAT-only criteria could be detrimental locally.

*Second, the authors rely on previous papers that they have published (Ribes et al, 2021) or submitted (Qasmi and Ribes, 2021) to present the methodology. While I don't have strong objections to this, it does mean that this paper is not as self-contained as it might be. This creates some challenges for readers who are interested in the methodological details since some of the details of the application of the method are different in this paper than in Ribes et al (2021) and Qasmi and Ribes (2021).*

This is a good point which was also raised by reviewer 2. The revised version of the manuscript will include a much more detailed description of the method to facilitate reading and make the paper more self-contained.

*Third, the paper does not provide very much information about the observed regional temperature time series and how it is computed. A hard to obtain paper by Mestre et al. (2013) is cited describing the homogenization software that is used in the production of the dataset, but apparently homogenizations is only applied to the early part of the station records that go into the French national thermal index – data from 1947 onwards are not homogenized. When I went searching for the French « Indice Thermique Nationale », I found it surprisingly difficult to find a description. I did, ultimately, find a map showing the locations of the 30 long-running stations that comprise this temperature indicator, and wasn't surprised to see that many of those stations appear to be located in or near urban centers. A potential concern about the data, therefore, is whether these stations are affected by the gradual expansion or intensification of urban heat islands, in which case one might worry that the recorded warming may not be representative of the warming experienced by Metropolitan (mainland) France as a whole. The paper hints that declining aerosol burdens may have something to do with the rapid warming in the region, but it may also be the case that the data are not as homogeneous as thought.*

We have implemented two additional tests to assess the robustness of our observed data in terms of the long-term warming trend.

The first test was designed to assess the potential influence of urban heat islands on long-term trends. As pointed out by the reviewer, our data (the "Indicateur Thermique National") is constructed as the average of 30 long-running stations. These stations are often located in urban areas. The average density of inhabitants in the corresponding towns is about 4000p/km². We therefore selected 30 alternative long-running stations, in less dense areas. Each of the alternative stations is close to one of the original ones, so that the geographical representativeness is kept roughly unchanged. The temperature series at these alternative stations were homogenised over the full observation period. So, we can make a comparison of our original series with a fully homogenised alternative that is much less likely to be affected by the urban heat island effect. The results are shown in the Figure R1 below. The

long-term warming trend diagnosed by this alternative dataset is very close to that assessed from the original data, suggesting that there is no discernible heat island effect in our data.

[Figure]

Fig R1 : Comparison of the official « Indicateur Thermique National » used by Météo france and within this study ("Opertaional ITh", black), the average of the same 30 ground-stations using their homogenised monthly time-series ("Homogenized", blue), and an alternative reconstruction of mean temperature over France based on the average of 30 homogenised series located in less dense areas ("Homogen – No UHI", green).

As a second test, we computed an estimate of annual mean temperature over France from the well known CRUTEM5 dataset. This dataset provides homogenized estimates of temperature anomalies on a 5°x5° resolution worldwide. CRUTEM data are typically used (among other useful datasets) in IPCC assessments. A France crude averaged is obtained by weighting CRUTEM grid-boxes by the ratio of surface that falls within mainland France. We show the results in Figure R2 below. Again, this test suggests that the warming trend assessed from the Indicateur Thermique National is relatively robust. While some difference between the 2 datasets is discernible in the mid 20th century, the diagnosed 1900-1930 to 2000-2020 warming are very close : +1.54°C in IthN, vs +1.58°C in CRUTEM. So CRUTEM warms even slightly more than our data.

[Figure]

Fig R2 : Comparison of the « Indicateur Thermique National » (black) with the reconstructed mean temperature over France as derived from the CRUTEM5 dataset (red).

These two tests support the reliability of the national index. As a consequence, we still use this dataset as the main observed data over mainland France. However, we have revised our manuscript in two ways:  we have mentioned the closeness with CRUTEM in the main text (section 2.2), and we provide more discussion on the observational uncertainty in our last section (discussion).

*Finally, I think it should be recognized that the technique apparently does not account for uncertainty in covariance and mean vector estimates that enter into calculation of the posterior distribution for regional mean temperatures. To do so, wouldn't it be necessary to use a more complex hierarchical Bayes model that uses priors on these parameters to describe their potential uncertainty?*

This has been recognized as a final remark in the method description section.

*Some specific comments:*

*72-73: This sounds like it might be unnecessarily complex. Since the interest is just in the Metropolitan France spatial mean temperature, does this interpolation scheme simply amount to weighting each climate model grid box by the fraction of the box that lies within Metropolitan France? Is there perhaps a step here that is not described, such as using only the 30 10-km grid boxes that contain the 30 stations that contribute to the ITh to calculate the model version of the ITh?*

Our goal in applying this treatment is to approximate the average temperature over mainland France. Similarly, we assume that the ITh will provide a good approximation of that France-wide average (note: similar assumption is being made for GSAT, despite incomplete spatial coverage).
In practice, we agree that using a 10-km resolution is probably not required, but since it remains easy to compute, we see no issue here. The result from our procedure will, indeed, be very close to the direct weighting of model grid bow that is proposed by the referee, with just one noticeable difference: model grid points with a land-sea fraction below 75% are excluded. In this way, the calculated France-average is not contaminated by SSTs (that are known to warm less than neighbouring land). Since, in our view, this is a desirable property, we did not revise our calculation.

*82: It does seem a bit ironic that, on the one hand, the authors repeat concerns about pattern scaling (lines 42-47), but then rely on it quite heavily throughout the rest of the paper (note that I personally would do so as well).*

In our view, there is no contradiction between l42-47 and pattern scaling. Indeed, the concerns expressed in l42-47 remain valid even if the pattern scaling assumption applies. First, combining pattern uncertainty with uncertainty in the amount of GSAT warming at a given date is challenging. Second, the warming pattern shown in IPCC-style maps of warming for a given GWL are multi-model average, which ignores observational information. In l42-47, we just suggest that better estimates **could** be proposed, even under the pattern scaling assumption.

*99: Is the fact that the 30 stations are all at low elevation a limitation?*

Yes, potentially, as it was shown that warming over western Europe is larger at higher rather than lower elevation. A quick calculation suggests that this effect is limited, though. So we just mention this as a potential limitation, without further analysis.

*120-126: I was initially misled by the structure of equation (1) and the explanation of the matrix H. It some time for me to realize that H's function was simply to select a piece of the vector x (i.e., it consists of zeros, except for one or more embedded identity sub-matrices that live within H). Describing the methodology more completely within the paper so that one*

*doesn't have to go searching in related papers would greatly help the reader. This could be done in the supplement, for example, which would probably be better used for that purpose than for dealing quickly with precipitation in any case).*

We have added much explanation about the method in the revised version of the manuscript – this was also asked by Reviewer 2. In terms of format, we've added some explanations in the main text, and added some specific aspects in the new Appendix B.

*154: I'm not sure that "measurement uncertainty" would be quite the right description of the information that is represented by the spread of the HadCRUT5 ensemble. Instrumental measure error might be part of that spread, but uncertainty associated with incomplete instrumental coverage and missing data, the adjustment of different types of SST data, the calculation of local anomalies, etc, all contribute, do they not? What we learn, I think, is something about the uncertainty of the estimated global mean temperature anomaly for each year.*

*158-160: In this case, I think it is appropriate to talk about measurement error, and to assume that it is small relative to internal variability (provided that the stations have very little if any missing data).*

We agree with these comments. We added the following sentence to the revised version: *Note that measurement error is meant in a broad sense here, as it encompasses all errors involved in estimating a global or regional temperature average, including individual measurement error, but also the treatment of incomplete data coverage, homogenization uncertainty, and others.*
However, we keep the "measurement error" terminology for simplicity. We think this is defensible wording, as observations can be considered as a "GSAT measurement", in which errors include all sources mentioned by the reviewer.

*242-255: This seems to be about attribution, and yet the next subsection has the title "Attribution". Section 3.2 could perhaps be better entitled "Contributions to warming from different forcing agents" or some such.*

We have revised the title of Section 3.2 into "Attribution to different forcing agents".

*295: The discussion of the climate normals adds an interesting twist to the paper, but before making a lot of the apparent lack of uniformity of warming and discussing evidence for a change of seasonality, shouldn't one add uncertainty bands to Figure 4?*

This is a very good point. Assessing the uncertainty on climate normals, and how they warm, is indeed of interest. In the revised version, we have introduced an original bootstrap approach to assess uncertainty in climate normals, leading to a revised (and improved) Figure 4. Uncertainty analysis reveals that uncertainty in daily warming estimates is pretty large, typically +/-.5°C. As a result, observations alone do not seem to provide clear evidence for a seasonal-dependent warming, so far. We revised (i.e., softened) our

comments accordingly. Still, the summer to winter warming ratio as estimated from observations is very close to what models simulate in the long-term.

*332: It would be good to add a reference to a paper that describes and analyzes this "well-known" enhancement.*

We have added some references:
- Giorgi & Lionello (2009) Climate change projections for the Mediterranean region
- Terray and Boé (2013) Quantifying 21st-century France climate change and related uncertainties
- Lionello & Scarascia (2018) The relation between climate change in the Mediterranean region and global warming.

*464-469: Should one also ask a question about the observations? Using a classical D&A technique, Sun et al. (2016, doi: 10.1038/NCLIMATE2956), showed that about 1/3 rd of the recorded warming in China between 1951 and 2010 could be attributed to urban warming, and that the remaining warming of about 1°C was much more consistent with the global land mean warming over that period.*

Agreed. We did add a question about observations, including an explicit mention of the heat island effect.